Research




**Subject Areas:**
materials science

**Keywords:**
layered double hydroxides, SiO$_2$, composites, mechanical properties

**Author for correspondence:**
Marija M. Vuksanović
e-mail: mdimitrijevic09@gmail.com

This article has been edited by the Royal Society of Chemistry, including the commissioning, peer review process and editorial aspects up to the point of acceptance.


ROYAL SOCIETY OF CHEMISTRY


# Inorganically modified particles FeAl-LDH@SiO$_2$ as reinforcement in poly (methyl) methacrylate matrix composite

Marija M. Vuksanović[1], Adela Egelja[1], Tanja Barudžija[2], Nataša Tomić[3], Miloš Petrović[4], Aleksandar Marinković[4], Vesna Radojević[4] and Radmila Jančić Heinemann[4]

[1]Department of Chemical Dynamics and Permanent Education, 'VINČA' Institute of Nuclear Sciences - National Institute of the Republic of Serbia, and [2]Department of Theoretical Physics and Condensed Matter Physic, 'VINČA' Institute of Nuclear Sciences - National Institute of the Republic of Serbia, University of Belgrade, Mike Petrovića Alasa 12-14, 11351 Belgrade, Serbia
[3]Innovation Centre of Faculty of Technology and Metallurgy in Belgrade, Karnegijeva 4, 11120 Belgrade, Serbia
[4]Faculty of Technology and Metallurgy, University of Belgrade, Karnegijeva 4, 11120 Belgrade, Serbia

MMV, 0000-0003-1872-195X; MP, 0000-0002-2460-7793

Silica particles were obtained from rice husk to which layered double hydroxide particles were deposited (weight ratio 1 : 1). $Fe^{2+}$-$Al^{3+}$ layered double hydroxides (FeAl-LDH) were synthesized by co-precipitation with ratios Fe : Al of 3 : 1 in the presence of SiO$_2$ particles from the rice husk. Characterization of the synthesized FeAl-LDH@SiO$_2$ particles was performed by X-ray diffraction, Fourier transforms infrared spectroscopy (FTIR) and scanning electron microscopy with EDS. Prepared FeAl-LDH@SiO$_2$ particles were used as reinforcing agents in 1, 3 and 5 wt% quantity in poly (methyl) methacrylate matrix. The aim of this study was to examine whether FeAl-LDH@SiO$_2$ particles affect the mechanical properties of polymer composite materials. The morphology of the composites was examined using a field emission scanning electron microscope. Microindentation, tensile and impact testing determined the mechanical properties of the obtained composites.

## 1. Introduction

Two-dimensional materials gained a lot of importance with the development of nanomaterials. They are nanosized in one

dimension and have layers of activated material able to establish contact with the environment or materials in their vicinity. Layered double hydroxides (LDH) are layered compounds represented by a general formula represented by a general formula ($M^{2+}{}_{1-x}$ $M^{3+}{}_{x}$ $(OH)_2$)($A^{n-}{}_{x/n}$ $\cdot mH_2O$), where $M^{2+}$ ($Mg^{2+}$, $Cu^{2+}$, $Ni^{2+}$, $Zn^{2+}$, $Cd^{2+}$, $Co^{2+}$) and $M^{3+}$ ($Fe^{3+}$, $Al^{3+}$, $Ga^{3+}$, $Cr^{3+}$, $^{+}$, $Mn^{3+}$) are divalent and trivalent metal cations, and $A^{n-}$ is an interlayer anion. In these hydroxide layers, some of the $M^{2+}$ cations are isomorphy substituted with $M^{3+}$, thereby generating a positive charge. The higher (excess of) charge is compensated by the hydrated anions located in the interlayer gallery. These layered configurations give them the ability to be used extensively as anion-exchanging materials (adsorption, polymerization, catalysis, photochemistry, electrochemistry, biomedical and environmental application, etc). LDH also has the ability of cation exchange, which has been the subject of many studies in recent years (photocatalysis…) [1,2]. In LDH, each metal cation is octahedrally coordinated by OH- and these OH- ions surround the space with radius of 0.07 nm. Therefore, metal cations with radius not far from 0.07 nm can be incorporated into LDH. In the case of larger ions ($Mn^{2+}$, $Pb^{2+}$, $Cd^{2+}$, $Ca^{2+}$, $Y^{3+}$, $La^{3+}$) incorporation, the close-stacking configuration will be distorted to some extent. It was recently confirmed that LDH compounds, previously thought to be capable only of anion exchange, are capable of cation exchange, opening a new perspective to use those materials [3]. The interlayer space is filled with exchangeable inorganic (carbonates, chlorides, nitrates, sulfates, complex anions etc.) or organic anions (carboxylates, allylsulfides, glycerolate, polymeric anions, biochemical anions etc.) along with water molecules [4].

By introducing LDH fillers into the polymer matrix [2], a composite material of different functions and structures can be obtained, and the role of LDH can be to change or improve the functionality of the polymer matrix [5]. Composites made with different combinations of polymers and LDH fillers show superior physico-chemical properties and have wide application in the fields of light, electricity, magnetism and chemistry [6]. Depending on the composition of LDH, their dispersion efficiency and orientation can be used as functional or structural reinforcements in polymer composite materials [7–9]. Most studies use LDHs containing Mg/Al or Zn/Al; however, other combinations of metal cations can also yield interesting mechanical properties of composite materials such as the influence of Ni/Al, Zn/Al and Co/Al-LDH on mechanical properties of poly (methyl) methacrylate (PMMA) [10]. In LDHs, interlayer anions serve a dual purpose. On the one hand, they contribute to the LDHs' layered structure, increasing interlayer distance; on the other hand, they promote compatibility with the polymer matrix [11].

Combinations of organic and inorganic components of composites have attracted attention due to their application in various fields of materials such as plastic reinforcement, biomaterials, which can be used as sensors, etc. [12–15]. The use of these composites is becoming increasingly popular in the restoration of teeth, where they replace metal dental amalgams, and the reason for that is they are more aesthetically pleasing, have better mechanical properties and are easy to make [16–20]. Silica is the most common reinforcement in dental composite materials [21–23]. Composites containing silica have good compatibility with the tissue, they open a great possibility to be shaped in a defined form and their colour can be adjusted to fit many aesthetic demands. Silica particles have become widespread because they increase the strength of reinforced polymer composites [24–26]. Silica nanoparticles have found application in biomedicine, due to their large surface area and pore volume, they facilitate the delivery of antimicrobial biomolecules in the PMMA matrix [27–29].

Composites have a tremendous possibility of using the best of all components that are incorporated in their preparation. Acrylic composites have many possible uses such as medicine, electronics and dentistry. On the other hand, the main problem with acrylates is their limited toughness. The introduction of a ceramic phase provides an opportunity to improve the mechanical properties of composites such as hardness, modulus and polymerization shrinkage. The main problem is to create the right balance between those properties and the toughness of the material. Different forms of fillers were studied because of their possible improvement of polymer matrix materials [17,30,31]. It was shown that the crystal structure of alumina-based reinforcement enables a balance between the toughness and hardness of the material and to tune those properties. Alumina itself was modified using the addition of ferrous oxide [18,32]. The surface modification of the particles showed the possibility to improve the bulk properties as well as adhesion properties of acrylic films [19,33–35]. Other reinforcements were tested as the improvement for acrylic matrix improvements such as zirconia [36]. The main idea for this research was to use silica, from green sources, as reinforcement and to adjust the bond between the reinforcement and the matrix using the inorganic coating of the silica particles. Silica particles were produced from the rice husk, and they were of very fine dimensions enabling good prospective material for future use in composites [37,38]. The use of the co-

precipitated Fe/Al-LDH as the surface modifier on silica obtained from bioresources was a challenge for producing a new composite material having improved properties [39]. The silica particles are a very well-established reinforcement used in particle reinforced polymer composites. The interphase between the reinforcement and the matrix can improve material characteristics, and often the silane molecules are added to the active places on silica particles in order to establish good bonding between phases. The idea behind this modification was to modify the surface of silica particles using LDH that would further enable the building of bonds with the matrix. This is also useful to modify the tendency of silica particles to agglomerate and to enable better dispersion of particles in the composite. The aim of the research was to examine the use of so modified particles and to improve mechanical properties of the obtained material. The mechanical properties are to be studied and commented on in view of this sort of surface modification.

The silica particles used in this study are submicron (i.e. lower than microns but larger than 10 nm) [38]. The common ratios vary from 2 to 4 if the LDH structure is favoured. Other ratios are producing other phases and sometimes lead to the formation mostly of divalent ion oxides; in this case, it would produce the oxide phase of ferrous ion [40]. The obtained composite particles were characterized and used as reinforcements in PMMA-based composite materials. Composites with different quantities of FeAl-LDH@$SiO_2$ fillers 1, 3 and 5 wt% were prepared. The strength, hardness and toughness of the obtained composite materials were tested.

## 2. Material and methods

As the source of the PMMA matrix, the auto polymerizing set for denture base Acrylate – R (Laser Dental Products). Rice husks were obtained from a rice producer from Levidiagro, Kočani, North Macedonia. Sulfuric acid was used as received from Zorka Sabac. Precursors for LDH synthesis, $FeCl_2 \cdot 4H_2O$ (Merck-Alkaloid, Skoplje) and Locron L (Aluminium hydroxide chloride; $Al_2(OH)_5Cl \cdot 2{,}5\ H_2O$) were purchased from the Clariant company and were used to synthesize FeAl-LDHs; 1.0 mol $l^{-1}$ NaOH solution was used to adjust the pH solution in LDH synthesis.

### 2.1. Preparation of FeAl-LDH and FeAl-LDH@$SiO_2$

To remove the contaminants from the rice husk, it was first washed with water and then dried. A hundred milligrams of such dried shell was treated with 10 wt% sulfuric acid at 80° C for 3 h. Thereafter, the shell was filtered and washed with water until the pH reached 7. Once the acid was removed, the husks were dried at 50°C for 24 h. The obtained product was treated using the Bunsen flame to remove as much as possible of the organic part of the material. The obtained product was black containing the rest of the organic products. The final step was heat treatment at 800°C for 4 h in an oxidative atmosphere resulting in a white powder with no visible organic residues.

FeAl-LDH (molar ratio Fe : Al = 3 : 1) were synthesized by the method of co-precipitation from aqueous solutions under atmospheric conditions. $FeCl_2 \cdot 4H_2O$ (0.015 mol) and $Al_2(OH)_5Cl \cdot 2.5\ H_2O$ (0.005 mol) were dissolved in 100 ml of deionized water separately. Silica particles were added to the glass beaker, which was placed on a magnetic stirrer. Then, the aqueous solutions of $FeCl_2 \cdot 4H_2O$ and $Al_2(OH)_5Cl \cdot 2.5\ H_2O$ were prepared. The mass ratio of silica: LDH was 1 : 1. The aim of this synthesis is to form a precipitate layer (FeAl-LDH) on the present silica particles which are inert to the co-precipitation reaction; 1 mol $l^{-1}$ NaOH was added dropwise to the solution until the pH reached 10 when the experiment was stopped. The dispersion was allowed to stand for 24 h, then centrifuged at 6000 rpm for 10 min obtaining the centrifugal force that is equal to 2366 g. Particles were then washed with water until the pH of the effluent solution was neutral. The solid with filter paper was dried at 80°C for 24 h to give FeAl-LDH@$SiO_2$ particles. The procedure was selected for the batch production of particles in this study for the simplicity and the quantity of particles to be produced. The alternative procedure was using centrifugation and rinse of the product, and further ultrasonication and centrifugation could be an alternative to the procedure used [3].

### 2.2. Preparation of composites

As the source of the PMMA matrix, the auto polymerizing set for denture base preparation was used. All denture base materials consist of two components: one is the methyl methacrylate monomer and the powder is the polymerized PMMA. The powder that usually contains the initiator is dissolved in

the monomer and after that the rest of the monomer polymerizes. The material is declared to conform to the ISO 1567 class I type II. The producer suggested that the volume ratio of powder and liquid should be 2.5 : 1. The polymerization was done at room temperature. PMMA matrix and different amounts of obtained FeAl-LDH@$SiO_2$ particles were used for composite preparation. The obtained FeAl-LDH@$SiO_2$ particles were added to the liquid monomer component in a glass placed in an ultrasonic bath for 30 min to disperse the particles. Then, a powder (PMMA) component was added in a glass, and the mixture was mixed until a paste was obtained which was placed in a mould. A pure polymer PMMA sample was also prepared for comparison.

## 2.3. Material characterization

A Mira3 Tescan field emission scanning electron microscope (FE-SEM) operated at 20 kV was used to examine the morphology and size of the particles. A thin gold film had previously been applied to the samples. The morphology of the composite materials was observed using (FE-SEM), operated at 3 kV.

X-ray diffractometer (Ital Structure APD2000) was used to perform crystal phases of FeAl-LDH@$SiO_2$ in a Bragg–Brentano geometry using CuK radiation in the range from 5 to 80° 2$\theta$.

FTIR spectra of FeAl-LDH@$SiO_2$ fillers and composite materials using a Nicolet 6700 spectrometer (Thermo Scientific) in the attenuated total reflectance (ATR) were used. The spectra were ATR corrected co-additions of 64 scans at 4 $cm^{-1}$ spectral resolution. The OMNIC software was used on the Nicolet 6700 FT-IR spectrometer, which recorded spectra in the wavelength range of 4000 $cm^{-1}$ to 400 $cm^{-1}$.

The micro Vickers tester (Kleinharteprufer DURIMETI), with a rectangular diamond pyramidal indenter, was used to test the microhardness of the composite [41]. To obtain a reproducible VN value, the microhardness of the PMMA matrix and PMMA composites containing different amounts of FeAl-LDH@$SiO_2$ fillers was measured, using a load of 500 gf for 25 s. Three indents were performed at room temperature for each sample following ASTM E384-16 [42]. The Image-Pro Plus program was used to calculate diagonal lengths from images captured by the Carl Zeiss – Jena (NU2 optical microscope). The average diagonal results from the reported measurements were used to calculate microhardness using the following equation:

$$\mathrm{VHN} = 2\cos\frac{22^\circ P}{d^2} = \frac{1.8544P}{d^2}, \tag{2.1}$$

where $P$ (kgf) is the applied load and $d$ (mm) is the indentation diagonal length [43].

The hardness and reduced modulus were calculated using the Oliver & Pharr [44] corrective parameter for the ball indenter [45].

Unloading segments of load versus displacement curves were used to fit power functions of the following form:

$$P = \alpha \cdot (h - h_f)^m, \tag{2.2}$$

where $P$ represents load, $h$ displacement and $\alpha$, $h_f$ and $m$ are parameters of the fit that are determined by the least-squares fitting procedure.

The slope of each fit, i.e. stiffness $S = \mathrm{d}P/\mathrm{d}x$, was calculated at maximum displacement/load ($h_{\max}$, $P_{\max}$).

Contact depth $h_c$ was then determined as

$$h_c = h_{\max} - \varepsilon \cdot \frac{P_{\max}}{S}. \tag{2.3}$$

The value of ε was set to 0.75 since this value matches the paraboloid of the revolution/spherical indenters.

The projected area of the hardness impression $A$ for the spherical indenter of the diameter $d$ was calculated as

$$A = (d \cdot h_c - h_c^2) \cdot \pi, \tag{2.4}$$

where $d$ represents the diameter of the spherical indenter.

Finally, the hardness and reduced modulus were calculated using the equations below:

$$H = \frac{P_{\max}}{A} \tag{2.5}$$

and

$$E_r = \frac{S}{2} \cdot \sqrt{\frac{\pi}{A}}. \tag{2.6}$$

Texture Analyzer EZ LX, Shimadzu with a 500 N load cell and a spherical indenter with a 4 mm diameter performed the microindentation experiments. In each experiment, the load was set to continuously increase by 0.25 N $s^{-1}$ until it reached 10 N. The maximum load was maintained for 20 s and then gradually decreased by 0.25 N $s^{-1}$ until the full release. The measurement was done using the indenter having a round tip with a diameter of 6 mm. The machine enables the fine measurement of the force applied on the surface of the material and the measurements of the indentation depth, a similar process as in nanoindentation. The process is repeated on six positions at the surface of the specimen and the data obtained are the mean of six measurements.

Tensile testing was carried out at room temperature on an Instron M 1185 universal testing machine. For all tests, the crosshead speed was 0.5 mm $min^{-1}$. The tension test specimen had standard dimensions, with a cross-section in the testing area of 2 × 5 mm and a length of 5 cm. The obtained data were used to assess the composite's tensile properties.

The composites' impact energy tests were carried out on the Shimadzu, Japan-based high-speed puncture impact testing machine HYDROSHOT HITS-P10. Total absorbed energy was automatically calculated following the load-time diagram, providing energy values for the maximum load and puncture point. The puncture point was defined as the point at which the force becomes zero. According to previous experience, the striker with a hemispherical head (diameter 12.7 mm) was loaded with the programmable impact velocity set at 1 m $s^{-1}$ [46]. The sampling time was 10 s, and 12 000 points were collected, some before, most during and after the impact. The striker was programmed to puncture all of the samples under consideration. The measurement was performed on three specimens from each group, and the results were presented as the mean values of the three tests.

# 3. Results and discussion

Figure 1 shows the morphology of synthesized silica particles and those with deposited LDH on the surface as revealed by FE-SEM analysis of the synthesized particles.

Silica particles present a very fine submicron structure as was expected from particles originating from plants (figure 1*a*). FeAl-LDH particles (figure 1*b*) when precipitated have mostly the morphology of small hexagonal plates that are obtained at the end of synthesis. The LDH layer is deposited on silica particles in a relatively thin layer, and the morphology of composite particles FeAl-LDH@$SiO_2$ is similar to that of the original silica particles. Very few hexagonal particles that are present in the image is the result of the precipitation of the excess of the FeAl-LDHs (figure 1*c*).

The EDS analysis was performed on a large surface area of the SEM sample to obtain the mapping image of the elemental composition of $SiO_2$ on which FeAl-LDH particles were deposited. The result of EDS analysis of silica particles on which FeAl-LDH particles are deposited is presented, demonstrating the elements of the composition of silica with FeAl-LDH particles, such as silicon, oxygen, iron and aluminium, table 1.

The synthesis of FeAl-LDH was done to obtain the molar ratio Fe:Al = 3 : 1, which was confirmed by EDS analysis according to atomic %. The Fe/Al/Si ratio confirms that the majority of LDHs are found on the surface of silica particles. This confirmed that the synthesis of silica particles to which FeAl-LDH particles were deposited on silica particles was successful. Cations like $Al^{3+}$ and $Fe^{2+}$ were chosen as the most promising ones for use in reinforced composites. It was proven possible that by tailoring the chemical composition of reinforcement, mechanical properties improve significantly [18,33]. Among the main reasons for such behaviour is the compatibility with the polymer matrix due to different amounts of available surface hydroxide groups.

Research regarding the preparation of Fe(II)–Al layered double hydroxides revealed that ratios 2 : 1 (LDH-2) and 3 : 1 (LDH-3) gave materials with a similar crystalline structure [47]. SEM analysis of the aforementioned LDH structures showed that LDH-3 possessed more compact structure/morphology than LDH-2. The loose structure of LDH-2 can be considered the most desirable one for the adsorption application since it is increasing the surface/contact area. But the structure of LDH-2 is not desirable for reinforced composites. The compact structure of reinforcement, such as LDH-3, will lead to improved mechanical properties of PMMA composites [46]. In addition, a slightly higher amount of available hydroxyl groups of LDH-3 than LDH-2 observed in FTIR spectra indicates the possibility of establishment of better interconnection with the polymer matrix. Therefore, the ratio 3 : 1 was considered to be the optimal one for better compatibility with PMMA and the LDH-3 structure for emphasized reinforcing effect.

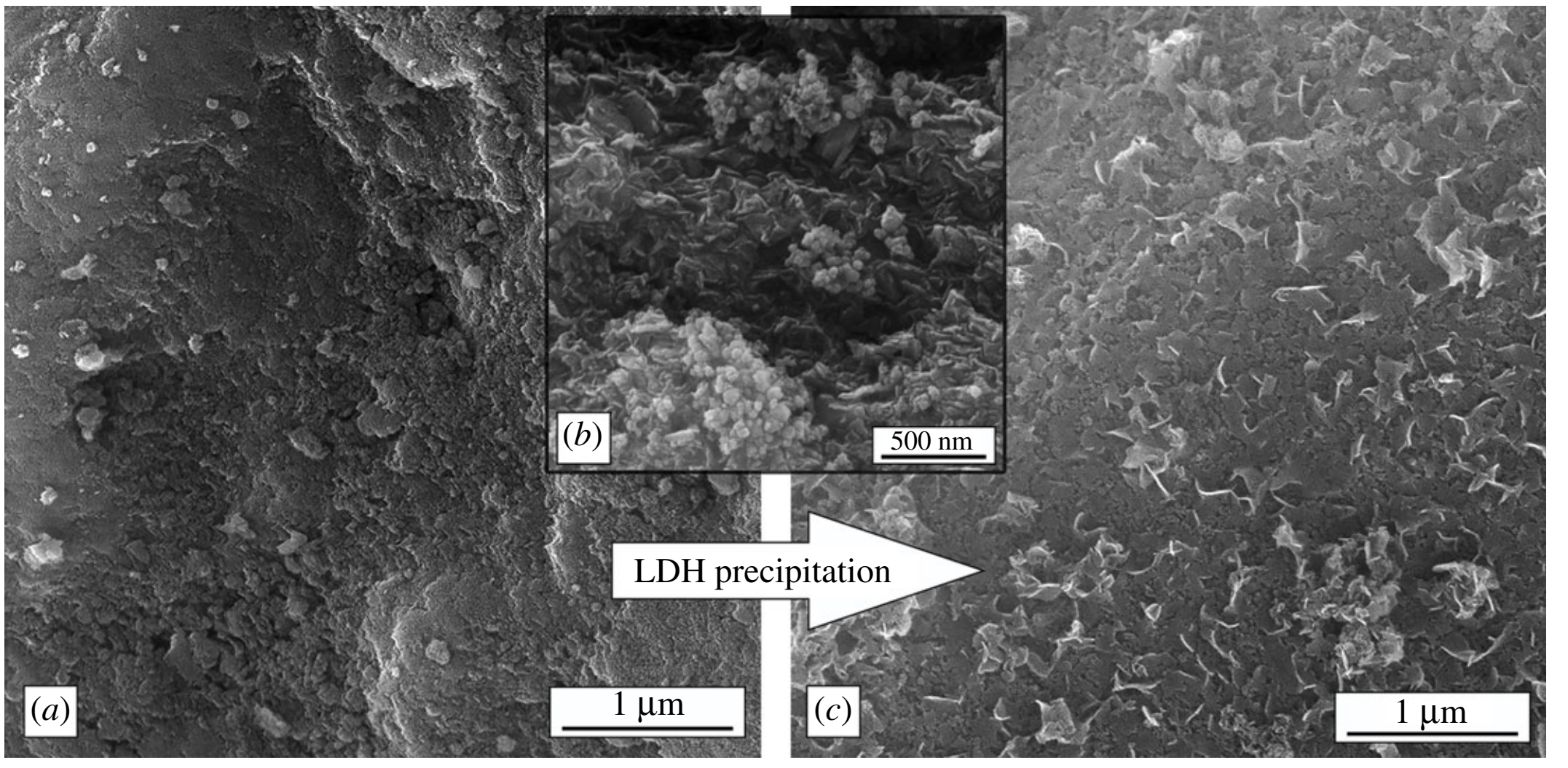


**Figure 1.** FE-SEM micrograph of fillers (*a*) $SiO_2$ particles, (*b*) FeAl-LDH and (*c*) FeAl-LDH@$SiO_2$.

**Table 1.** EDS for FeAl-LDH@$SiO_2$ particles.

| element | elements (%) | atomic (%) |
|---|---|---|
| O | 56.56 | 71.05 |
| Si | 27.14 | 19.74 |
| Fe | 14.07 | 5.44 |
| Al | 2.23 | 1.77 |
| sum | 100.00 | 100.00 |

The results from the EDS are suggesting that the layer of deposited LDH has the ratio of 3/1 of ions used in the preparation of the modifying layer. This crystal structure does not build the crystals that are easily recognizable from the XRD pattern. From available analysis, it is possible to establish the layer of the inorganic material on the surface of the silica particles having a predetermined ionic ratio. The structure was only observed using the SEM microscopy and some features conform to features observed in [48] where a better imaging technique was used. The ratio of silica to LDH could be estimated from the EDS analysis and the result says that the atomic ratio could be considered as Si/Fe/Al = 11.1/3/1.

The XRD patterns of $SiO_2$, FeAl-LDH and FeAl-LDH@$SiO_2$ fillers are shown in figure 2.

The XRD spectrum of $SiO_2$ (figure 2) does not have a defined crystal structure, so the obtained material is a non-crystalline very fine powder. The $SiO_2$ diffraction pattern shows a reflection characteristic of amorphous silica [45]. The XRD data from figure 3*b* reveals peaks that are characteristic of the LDH structure. The diffraction peaks at 11.7°, 23.7°, 39.4° and 63.1° corresponding to planes (003), (006), (015) and (113) of the typical layered structure. The XRD results showed that $FeCl_2$ and $Al_2(OH)_5Cl$ were successfully transformed into the LDH phase [48,49]. The basal spacing corresponding to the (003) reflection was found to be approximately 7.34 Å. The cell parameter $c$ calculated from $c = 3d_{003}$ is 2.202 nm. In pattern FeAl-LDH@$SiO_2$ from figure 2, there are no characteristic peaks of LDH structure because FeAl-LDH was synthesized in the presence of amorphous $SiO_2$ and broad peaks of $SiO_2$ obscured them. The XRD of composite FeAl-LDH@$SiO_2$ particles confirms that the crystal structure remains non-crystalline having only traces of the peaks of the LDH structure. The composite particles have the core of $SiO_2$ and a non-crystalline layer of LDH as the modifier on the surface. The absence of the crystal planes in the composite particles could be interpreted as a good step in the production of surface-modified silica particles for use in composite materials.

The chemical structure of the pure polymer matrix PMMA and composites with FeAl-LDH@$SiO_2$ fillers was also examined by FT-IR spectroscopy, figure 3.

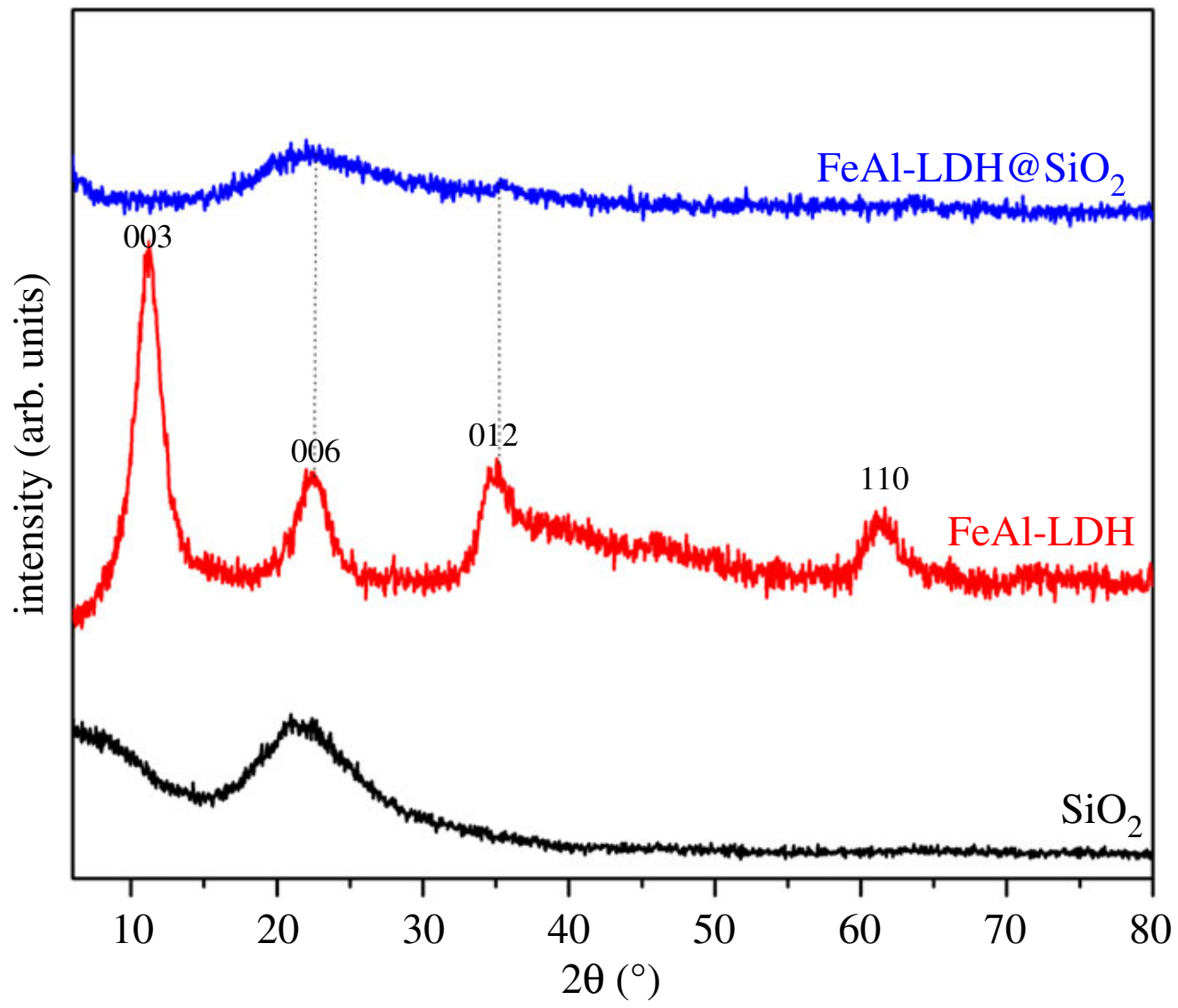


**Figure 2.** XRD patterns of $SiO_2$, FeAl-LDH and FeAl-LDH@$SiO_2$ fillers.

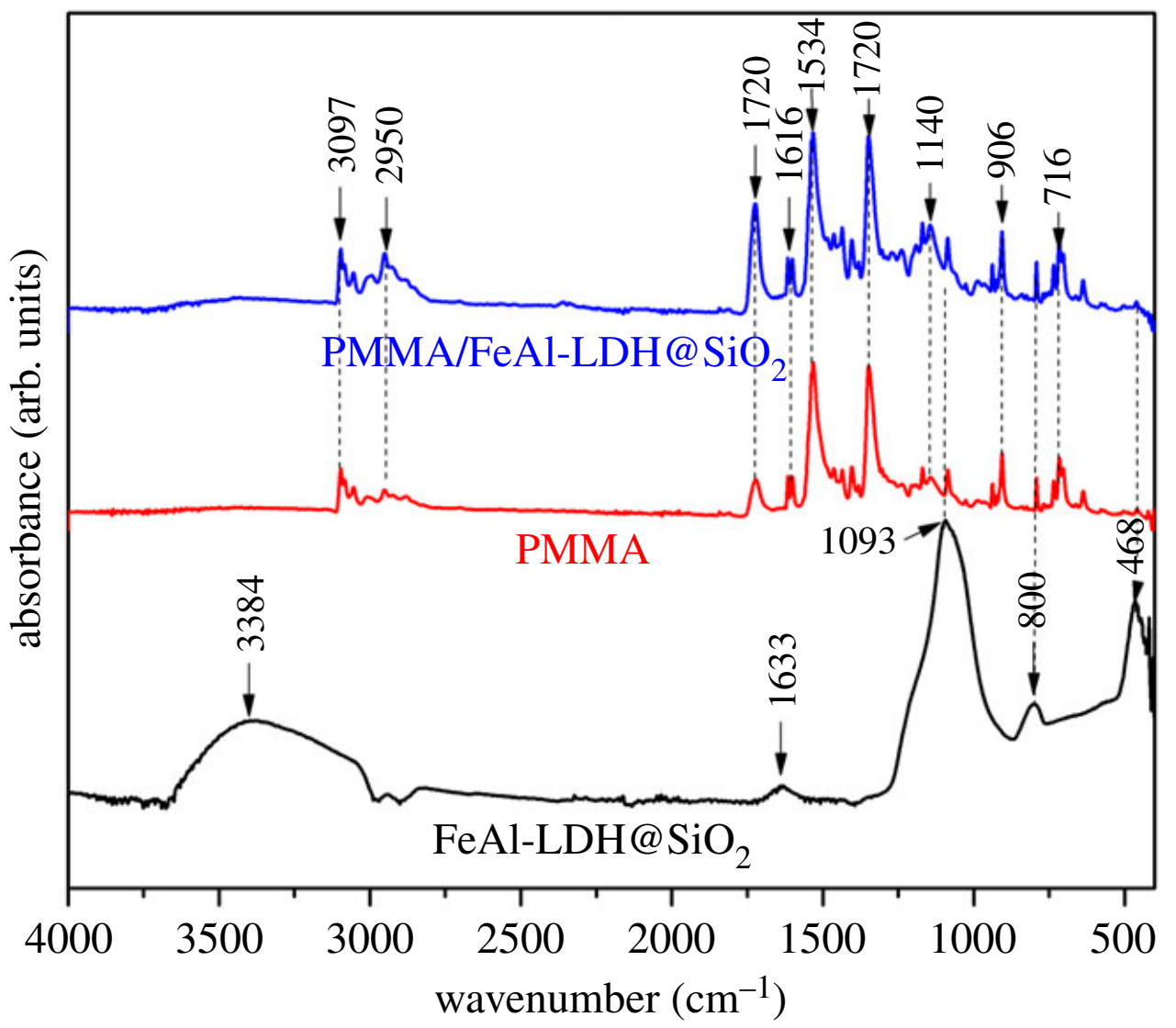


**Figure 3.** FTIR of FeAl-LDH@$SiO_2$, pure matrix and composite material with 5 wt% of FeAl-LDH@$SiO_2$.

The FTIR spectra show absorption bands caused by stretching and in-plane bending vibrations of the OH group of molecular $H_2O$ at around 3384 $cm^{-1}$ and 1633 $cm^{-1}$, respectively, which correspond to the O-H present on the FeAl-LDH lattice and free water molecules within the inter-gallery space [50–52]. Absorption bands at 1093 $cm^{-1}$, 800 $cm^{-1}$ and 468 $cm^{-1}$ are characteristic of silica particles. The absorption bands at 1093 $cm^{-1}$ and 804 $cm^{-1}$ show antisymmetric and symmetrical tensile vibrations of Si–O–Si. The absorption band at 468 $cm^{-1}$ corresponds to the bending vibration of the Si - O - Si bond, confirming the presence of silica [52]. The bands characteristic for silica particles also appears in the FTIR spectrum for composite material. PMMA has characteristic bands at 1720 $cm^{-1}$ v (C=O) and 1420 $cm^{-1}$ v (C=O) (C–O). The bands at 3097 and 2950 $cm^{-1}$ correspond to the methyl group's C–H elongation (CH3). The vibrations of the ester group C–O correspond to the band at 1140 $cm^{-1}$, while the elongation ranges C–C are at 906 and 716 $cm^{-1}$, respectively, [53]. The C=C stretching vibration of an aromatic was assigned the band at 1616 $cm^{-1}$ [48]. The bands at 1534 $cm^{-1}$ belong to $CO_3^{2-}$ in the interlayer space [53]. The chemical structure of silica particles, which was obtained from rice husk, and on which FeAl-LDH was deposited, was confirmed by FTIR spectroscopy.

Figure 4 shows micrographs of the fractured surface of a pure matrix and composite materials having 1, 3 and 5 wt% FeAl-LDH@$SiO_2$ particles.

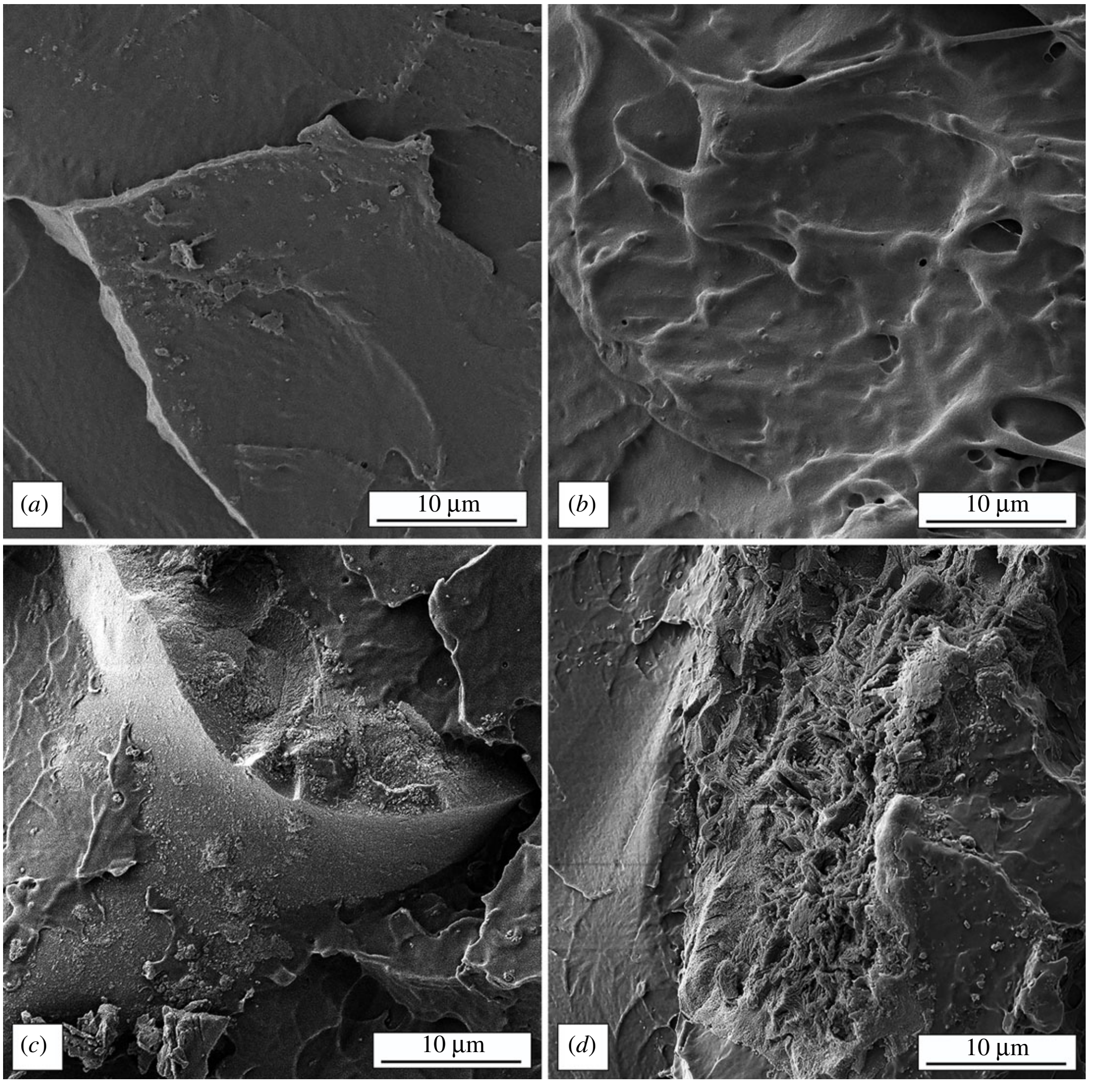


**Figure 4.** SEM micrograph of (*a*) pure matrix and composites with (*b*) 1 wt%, (*c*) 3 wt% and (*d*) 5 wt% of FeAl-LDH@$SiO_2$ fillers.

FE-SEM images of the PMMA matrix showed sharp lines on the fracture surface, suggesting a brittle fracture of homogeneous material [54]. The dispersion and interconnection of FeAl-LDH@$SiO_2$ layers in the PMMA matrix are critical for the properties of the synthesized composite. The FeAl-LDH@$SiO_2$ particles are randomly dispersed in the PMMA matrix, as seen in images of the composite's fracture surface. A close examination reveals lines that resemble wave-type wrinkles (due to the existence of LDH in the polymer matrix). By adding more filler, the FeAl-LDH@$SiO_2$ particles become aggregates around which the polymer is wrapped [55]. During the preparation of composites, it was observed that the reinforcements are easily added to the mixture. The ease of preparation through the mixing of the reinforcement and the matrix is better and the handling of the composite during the formation was easier [18]. The obtained images are from the impact testing samples after the testing and they reflect that the composite material is less brittle than expected. Adding the FeAl-LDH@$SiO_2$ particles enables easier handling of the material and better performance in high impact testing that is going to be discussed later.

The results of the Vickers microhardness of the pure matrix and composite materials are shown in figure 5.

The microhardness of pure matrix material was 0.162 GPa. Vickers microhardness of composites with 1 wt% of FeAl-LDH@$SiO_2$ fillers increases by 12.5% compared to the PMMA matrix. Composites with 3 wt% and 5 wt% FeAl-LDH@$SiO_2$ fillers have an increase in microhardness by Vickers of 16.7% and 31.2%, compared to the pure matrix material, respectively.

Results of micro indentation are presented in figure 6 as load–depth curves, while the calculated hardness and reduced modulus values are presented in table 2.

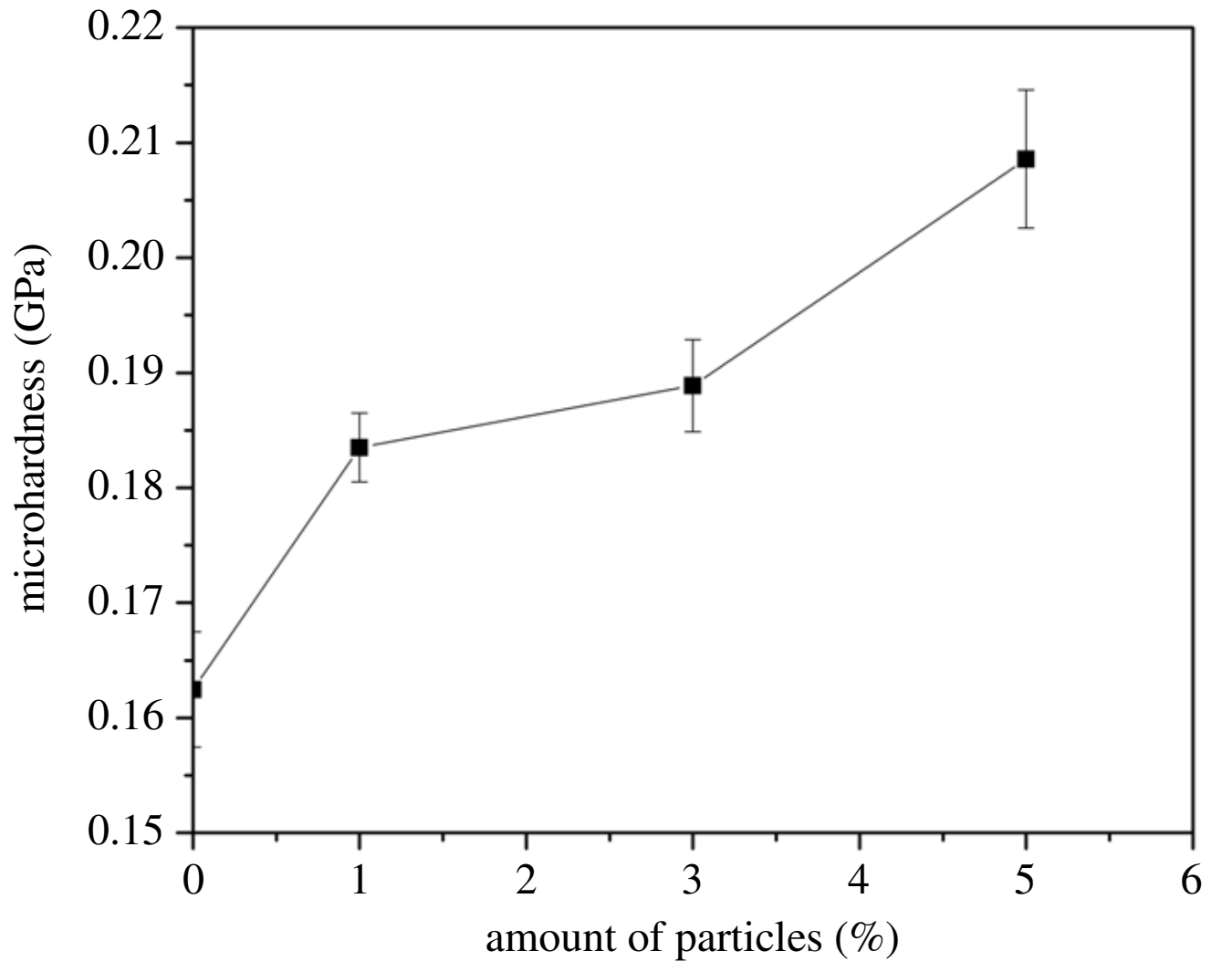


**Figure 5.** Results of measurements of the Vickers microhardness of composites.

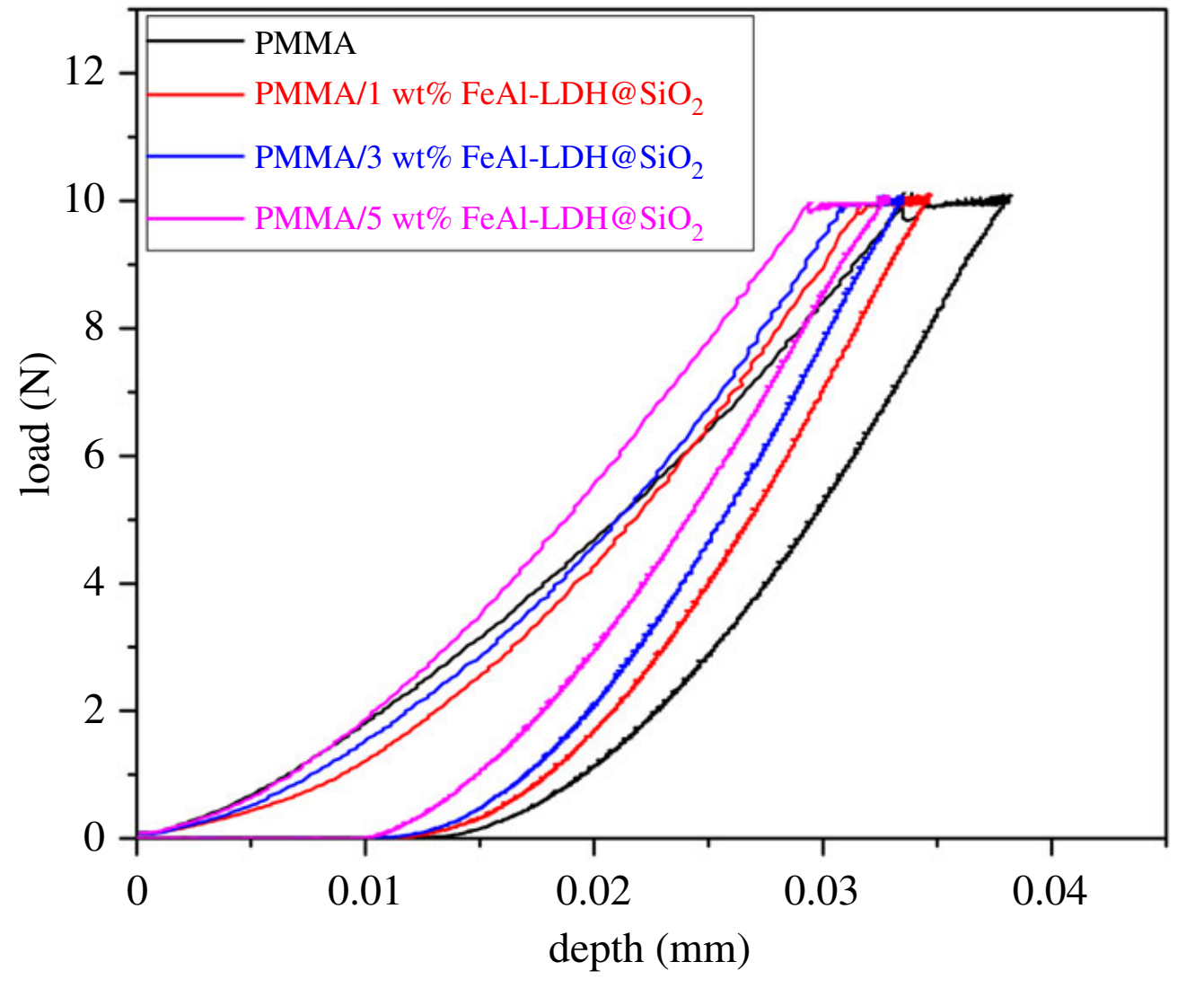


**Figure 6.** Loading-depth curves from the microindentation test for representative specimens of FeAl-LDH@$SiO_2$ reinforced composites.

**Table 2.** Results of microindentation, hardness and reduced modulus.

| sample | H (GPa) | Er (GPa) |
|---|---|---|
| PMMA | 0.032 ± 0.044 | 1.027 ± 0.25 |
| PMMA/1 wt% FeAl-LDH@$SiO_2$ | 0.033 ± 0.025 | 1.126 ± 0.18 |
| PMMA/3 wt% FeAl-LDH@$SiO_2$ | 0.039 ± 0.015 | 1.211 ± 0.20 |
| PMMA/5 wt% FeAl-LDH@$SiO_2$ | 0.041 ± 0.045 | 1.209 ± 0.13 |

The hardness obtained by the microindentation (Texture Analyzer EZ LX, Shimadzu) of composites with the addition of 3 wt% and 5 wt% of FeAl-LDH@$SiO_2$ fillers increased by 21.8% and 28.1%, respectively, compared to the pure matrix material. With the addition of 1 wt% of FeAl-LDH@$SiO_2$ filler, the reduced modulus increases by 9.64%, while by 3 wt% of FeAl-LDH@$SiO_2$, it increases by 17.9%.

Tensile testing is used to determine the composite's Young's modulus and power, which is dependent on the amount of fillers used. The tensile strength and modulus of elasticity (E) values are shown in table 3.

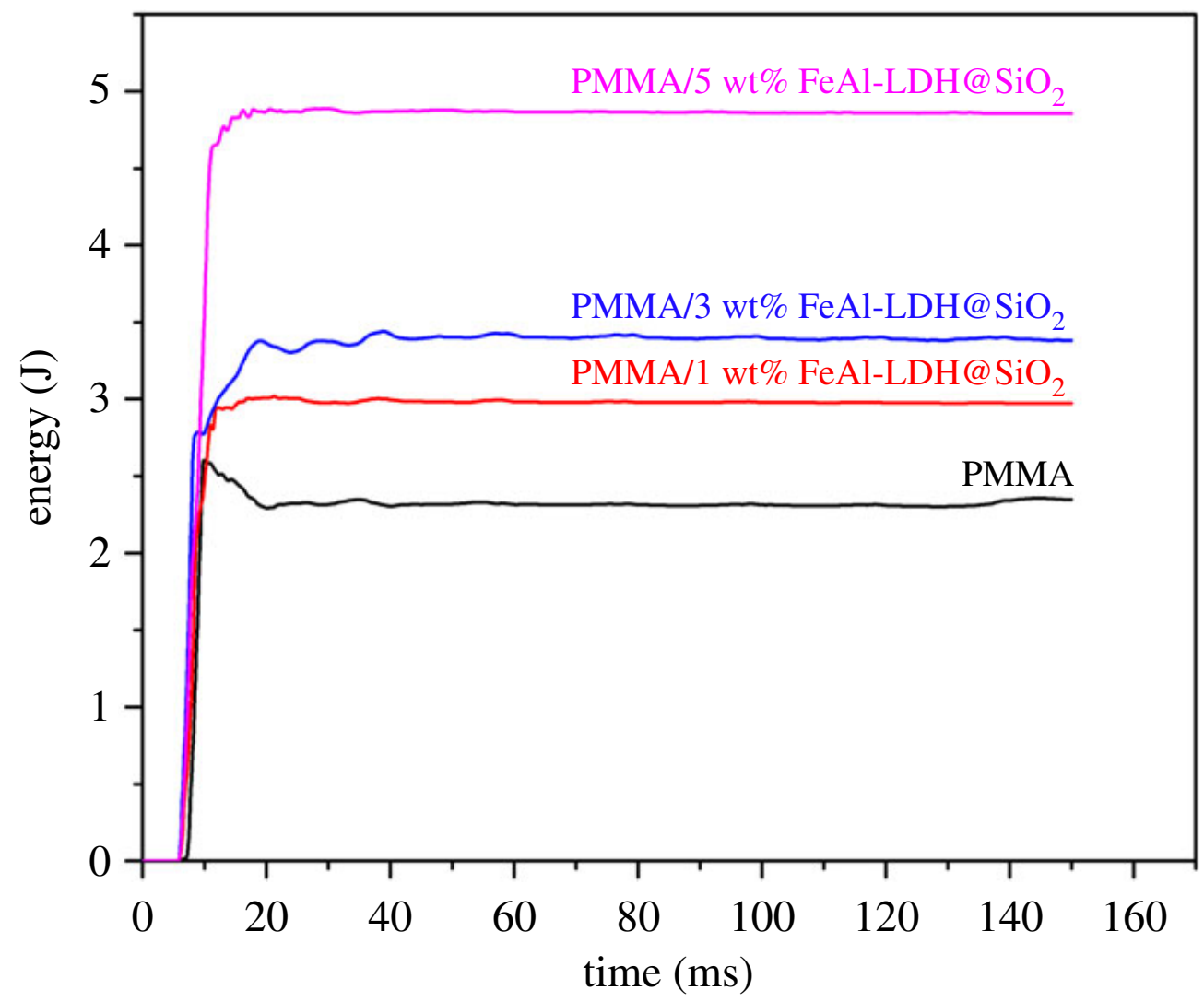


**Figure 7.** Energy absorption during impact testing of composites reinforced with FeAl-LDH@$SiO_2$ particles as a function of time and reinforcement content.

**Table 3.** Mechanical properties of composites were obtained using the quasi-static loading of the material under small deformation rates.

| samples | $\sigma$ (MPa) | $\varepsilon$ (mm mm$^{-1}$) | E (GPa) | K (MJ/m$^3$) |
|---|---|---|---|---|
| PMMA | 22.6 ± 0.029 | 0.0288 ± 0.016 | 1.122 ± 0.032 | 0.4659 ± 0.029 |
| PMMA/1 wt% FeAl-LDH@$SiO_2$ | 37.5 ± 0.049 | 0.0263 ± 0.035 | 1.248 ± 0.041 | 0.4321 ± 0.016 |
| PMMA/3 wt% FeAl-LDH@$SiO_2$ | 28.7 ± 0.048 | 0.0189 ± 0.006 | 1.365 ± 0.018 | 0.2448 ± 0.033 |
| PMMA/5 wt% FeAl-LDH@$SiO_2$ | 23.1 ± 0.021 | 0.0179 ± 0.005 | 1.377 ± 0.016 | 0.2150 ± 0.038 |

[a]$\sigma$—tensile strength, $\varepsilon$—deformation, E—Young's modulus of elasticity, K—adsorbed energy.

Increasing the amount of FeAl-LDH@$SiO_2$ fillers affects the reduction of the tensile strength of composite materials under the slow loading of the material [44]. According to the results, the Young's modulus of composite materials increases with increasing FeAl-LDH@$SiO_2$ filler content for 11.2%, 21.6% and 22.7% [17,56–59].

The resistance of the materials to high-speed impacts was tested on the high-speed puncture impact testing machine, (figure 7) to determine the toughness of composite materials. Table 4 compares the results of a study on the influence of reinforcement on the mechanical properties of composite materials.

The results of high-speed puncture impact testing indicate that the addition of the FeAl-LDH@$SiO_2$ fillers is improving toughness as measured by the maximum force. Figure 7 presents the results of prepared samples' impact testing. The result of the impact testing is the improvement of toughness that is up to three times that of the pure matrix. The energy required to fracture the material increases with the content of the reinforcement.

The interesting feature observed in this research relates to the way the load was applied to the specimen. In quasi-static loading experiments, increasing the reinforcement content improved both hardness and modulus, but the material's ability (maximum deformation during the tensile test) to deform is limited and decreases with the addition of the reinforcement. This observation is in correlation with the behaviour of nano- and micro-silica reinforcement filled epoxy [60]. The energy absorbed was calculated as the surface under the stress–strain curve obtained from quasi-static experiment. The energy absorbed before breaking the material decreased with the addition of the reinforcement (table 3).

The load increase in time is one of the factors influencing the crack propagation. This is one of the main factors that controls the crack propagation in the particulate composite material [61]. The specimens were tested using the controlled energy impact test, where the energy is suddenly transmitted to a material. The data acquisition system enables us to follow the energy transmitted to

**Table 4.** Comparison of the mechanical properties of composite materials found in the literature.

| composites | reinforcement content (%) | microhardness (%) | tensile strength (%) | impact testing (%) | reference |
|---|---|---|---|---|---|
| PMMA/$Al_2O_3$ fibre | 1 | 1.72 | — | — | [55] |
| | 3 | 158.2 | — | — | |
| | 5 | 120.6 | — | — | |
| PMMA/DMI/ $Al_2O_3$ Fe particles | 1 | 13.6 | 37.5 | — | [44] |
| | 3 | 37.3 | −5.3 | — | |
| | 5 | 30.3 | −22.0 | — | |
| PMMA/$ZrO_2$ particles | 1 1.5 | 15.4 7.38 | — | −29.7 | [17,44] |
| | 3 3 | 24.1 13.4 | — | −26.2 | |
| | 5 5 | 32.8 19.9 | — | −9.5 | |
| | 7 | 23.3 | — | −28.8 | |
| | 10 | 30.1 | — | −41.4 | |
| PMMA/$TiO_2$-ZnO particles | 1 | 33.3 | — | — | [56] |
| | 2 | 56.9 | — | — | |
| | 3 | 92.5 | — | — | |
| | 4 | 141.9 | — | — | |
| | 5 | 226.9 | — | — | |
| PMMA/orange peels powder | 2 | — | 5.88 | 14.8 | [57] |
| | 4 | — | 9.80 | 32.4 | |
| | 6 | — | 3.92 | 48.6 | |
| | 8 | — | 1.96 | 44.6 | |
| PMMA/$SiO_2$ particles | 0.5 | 4.41 | −10.1 | — | [58] |
| | 1 | 12.1 | −14.0 | — | |
| | 5 | 17.9 | −40.9 | — | |
| PMMA/LDH particles | 5 | — | 52.8 | — | [59] |
| | 10 | — | 42.3 | — | |
| | 15 | — | 38.4 | — | |
| | 20 | — | 35.2 | — | |
| PMMA/LDH particles | 1 | 12.5 | 37.5 | 38.1 | [in this paper] |
| | 3 | 16.7 | 28.7 | 57.1 | |
| | 5 | 31.2 | 23.1 | 128.6 | |

the specimen in time. In figure 7, the energy transmitted to the specimen is presented for the specimens having 1%, 3% and 5% of reinforcement and those are compared to the pure matrix material. The energy required to break the material increased with the content of reinforcement meaning that the material is exhibiting better toughness under such conditions. This feature would be studied in further research to explain it in detail [62,63].

# 4. Conclusion

Silica particles were prepared from a green source—rice husks. The surface of silica particles was modified using the inorganic modifier based on FeAl-LDH deposited in the co-precipitation process. FeAl-LDH@$SiO_2$ composite particles resulted from this process. The structure of the obtained fillers was examined by the XRD method. The silica particles kept their non-crystalline structure, and the deposited layer was thin enough to remain non-crystalline too. The obtained composite fillers were

used to reinforce a PMMA matrix. Composite materials were made with a different amount of FeAl-LDH@$SiO_2$ composite fillers. The mechanical properties of the specimen were investigated using the Vickers microhardness tester and the addition of the reinforcement increases the hardness of the composite. The specimens were subjected to the static loading tensile test. Those tests revealed the relatively brittle nature of the material. The calculated energy absorbed before fracture in quasi-static loading tests reveals the relative brittleness of the material. The material's modulus and tensile strength increase, but the deformation preceding the fracture decreases. The microindentation was done under controlled conditions to obtain the modified hardness and modulus of elasticity values. The hardness and modulus increased with the addition of the reinforcement. Compared to Vickers microhardness that increased with the addition of the reinforcement measurement, results confirmed the increase of hardness and modulus of the material. The calculated energy absorbed in quasi-static testing decreased with the addition of the reinforcement.

Toughness was measured using the high-speed impact tester and the results indicate that substantial improvement is obtained by adding the reinforcement in the material. The toughness for the specimen having 5 wt% of FeAl-LDH@$SiO_2$ fillers is three times higher in energy absorbed in this impact testing compared to the matrix. Those results indicate that the material subjected to high-speed impact exhibits better behaviour when containing more reinforcement in the composition.

Data accessibility. The datasets supporting this article have been uploaded as part of the electronic supplementary material.

Authors' contributions. M.V. was involved in conceptualization, data curation and writing the original draft. A.E. was involved in investigation and formal analysis. T.B. was involved in formal analysis. N.T. was involved in methodology, visualization, writing the review and editing. M.P. was involved in software and methodology. A.M. was involved in resources and data curation. V.R. was involved in writing the review and editing, and methodology. R.J.H. was involved in supervision, writing the review and editing.

Competing interests. We declare we have no competing interests.

Funding. The research was funded by the Ministry of Education, Science and Technological Development of the Republic of Serbia (Contract nos. 451-03-68/2021-14/200135 and 451-03-68/2021-14/200287).

Acknowledgements. The authors would like to thank the Ministry of Education, Science and Technological Development of the Republic of Serbia (Contract nos. 451-03-68/2021-14/200135 and 451-03-68/2021-14/200287).

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
