## [Peer Review File · Royal Society Open Science]

Review History

RSOS-210835.R0 (Original submission)

Review form: Reviewer 1

Is the manuscript scientifically sound in its present form?

Yes

Are the interpretations and conclusions justified by the results?

No

Is the language acceptable?

Yes

Do you have any ethical concerns with this paper?

No

Have you any concerns about statistical analyses in this paper?

No

Recommendation?

Major revision is needed (please make suggestions in comments)

Comments to the Author(s)

Royal Society Open Science - Manuscript ID RSOS-210835

The manuscript reports the purification of rice husk silica, deposition of LDH on the silica surface and use these materials as filler into PMMA. The topic is interesting and deserves publication but many corrections need to be implemented in the text before acceptance.

Recommendations/changes/suggestions:

- 1- Pg. 1 - Line 46 - are lamellar compounds. Better to say: layered compounds
- 2- Pg. 1 - Line 47 - exchangeable anions. Recently it was reported that LDH can also exchange cations and even cations/anions simultaneously, please update this information.
- 3- Pg. 1 - Line 50 - In LDH, the metal ions must have an ionic radius similar to Mg^{2+} (0.65 Å). The interlayer anions are carbonates, chlorides, nitrates and sulfates. The first information is not correct and the second is incomplete, organic and inorganic anions can be intercalated, please correct.
- 4- Pg. 3 - Line 47 - f Fe-Al LDH. A reason should be given for choosing this composition and also for the 3:1 molar ratio, which is very uncommon.
- 5- Pg. 3 - Line 48 - remove the pollution. Better to say: remove the contaminants.
- 6- Pg. 3 - Line 56 - were synthesized by a co-precipitation method. This information is incomplete, there are many different co-precipitations procedures, please explain details of the procedure.
- 7- Pg. 3 - line 60 - 2.93 g of $FeCl_2 \cdot 4H_2O$ and 1.07 g of $Al_2(OH)_5Cl \cdot 2,5 H_2O$. In chemistry the amounts should be given in mols, not in masses. Please convert the numbers.
- 8- Pg. 4 - Line 4 - The suspension. The correct term is dispersion.
- 9- Pg. 4 - Line 5 - 6000 rpm for 10 minutes. The centrifugal force should be provided. Also, the filtering method is not a good method to avoid contaminants, better to centrifuge, remove the solution, dispersed the slurry with an ultrasound bath and repeated this procedure at least 5 times.
- 10- Pg. 4 - Line 18 - Figure 1 should be removed; it is too basic for this level of journal.
- 11- Pg. 4 - Line 20 - The set consisted of the prepolymerized powder and the monomer liquid. The complete information about these Chemical should be given to allow any reader reproduction.
- 12- Pg. 5 - Line 48 - Figure 2. FE-SEM micrographs of fillers a) SiO_2 particles, b) FeAl-LDH and c) FeAl-LDH@ SiO_2 . Where is Figure 2C????
- 13- Pg. 5 - All the figures need to be replaced due to the low quality.
- 14- Pg. 5 - It is recommended that all the discussion come after each figure otherwise it is difficult to follow in the text. Use the topic results and discussion
- 15- Pg. 7 - Line 4 - related to the SEM images, better magnifications would better support the comments in the text.
- 16- Pg. 7 - Line 12 - The Fe/Al/Si ratio confirms that the majority of LDHs are found on the surface of silica particles. How the authors explain the composition of the EDS if the ratio LDH: Silica was of 1:1? This proportion is far in the analysis and no comments are provided in the text.
- 17- Pg. 8 - Line 19 - The XRD diffraction peaks should be indicated in interplanar distances after the position of the peaks in 2 theta and the basal distance calculated by the higher order peak possible, never the first one. The sizes of the crystalline domains can be calculated by Scherrer equation, the a and b cell parameters from the 110 peak. Which is intercalated anion?
- 18- Pg. 8 - Line 22 - In pattern FeAl-LDH@ SiO_2 from figure 3 there are no characteristic peaks of LDH structure because FeAl-LDH was synthesized in the presence of amorphous SiO_2 and broad peaks of SiO_2 obscured them. This is not correct, if the proportion used is true, the

XRD pattern of LDH should appear in the mixture. This can be an indication that a very small amount of LDH was deposited. This information should be given in the text by using proper analysis since EDS is not quantitative.

19- Pg. 8 - Line 24 - crystal structure remains non crystalline. This is really strange, if crystals exist, the material is crystalline.

20- Pg. 8 - Line 27 - no crystalline layer of LDH. This is unlikely to be true, particles were observed with layered morphology, consequently LDH must be in the sample and crystalline material.

21- Pg. 8 - Line 33 - The absorption peaks. FTIR spectra preset bands, not peaks. Please correct all the text.

22- Pg. 8 - Line 47 - The FeAl-LDH@SiO₂ particles are randomly dispersed in the PMMA matrix, as seen in images of the composite's fracture surface. This is really to see in the images, a higher magnification and mapping the elements would give better information in this sense.

Review form: Reviewer 2 (Isa Emami Tabrizi)

Is the manuscript scientifically sound in its present form?

Yes

Are the interpretations and conclusions justified by the results?

Yes

Is the language acceptable?

No

Do you have any ethical concerns with this paper?

No

Have you any concerns about statistical analyses in this paper?

No

Recommendation?

Accept with minor revision (please list in comments)

Comments to the Author(s)

The manuscript can be published after considering all the comments in the attached file (see Appendix A).

Decision letter (RSOS-210835.R0)

Dear Dr Vuksanovic:

Title: Inorganically modified particles FeAl-LDH@SiO₂ as reinforcement in PMMA matrix composite

Manuscript ID: RSOS-210835

The editor assigned to your manuscript has now received comments from reviewers. We would like you to revise your paper in accordance with the referee and Subject Editor suggestions which can be found below (not including confidential reports to the Editor). Please note this decision does not guarantee eventual acceptance.

Please submit your revised paper before 04-Aug-2021. Please note that the revision deadline will expire at 00.00am on this date. If we do not hear from you within this time then it will be assumed that the paper has been withdrawn. In exceptional circumstances, extensions may be possible if agreed with the Editorial Office in advance. We do not allow multiple rounds of revision so we urge you to make every effort to fully address all of the comments at this stage. If deemed necessary by the Editors, your manuscript will be sent back to one or more of the original reviewers for assessment. If the original reviewers are not available we may invite new reviewers.

On behalf of the Subject Editor Professor Anthony Stace and the Associate Editor Professor Chaohua Cui.

RSC Associate Editor:
Comments to the Author:
(There are no comments.)

RSC Subject Editor:
 Comments to the Author:
 (There are no comments.)

Reviewers' Comments to Author:
 Reviewer: 1

Comments to the Author(s)

Royal Society Open Science - Manuscript ID RSOS-210835

The manuscript reports the purification of rice husk silica, deposition of LDH on the silica surface and use these materials as filler into PMMA. The topic is interesting and deserves publication but many corrections need to be implemented in the text before acceptance.

Recommendations/changes/suggestions:

- 1- Pg. 1 - Line 46 - are lamellar compounds. Better to say: layered compounds
- 2- Pg. 1 - Line 47 - exchangeable anions. Recently it was reported that LDH can also exchange cations and even cations/anions simultaneously, please update this information.
- 3- Pg. 1 - Line 50 - In LDH, the metal ions must have an ionic radius similar to Mg^{2+} (0.65 Å). The interlayer anions are carbonates, chlorides, nitrates and sulfates. The first information is not correct and the second is incomplete, organic and inorganic anions can be intercalated, please correct.
- 4- Pg. 3 - Line 47 - f Fe-Al LDH. A reason should be given for choosing this composition and also for the 3:1 molar ratio, which is very uncommon.
- 5- Pg. 3 - Line 48 - remove the pollution. Better to say: remove the contaminants.
- 6- Pg. 3 - Line 56 - were synthesized by a co-precipitation method. This information is incomplete, there are many different co-precipitations procedures, please explain details of the procedure.
- 7- Pg. 3 - line 60 - 2.93 g of $FeCl_2 \cdot 4H_2O$ and 1.07 g of $Al_2(OH)_5Cl \cdot 2.5 H_2O$. In chemistry the amounts should be given in mols, not in masses. Please convert the numbers.
- 8- Pg. 4 - Line 4 - The suspension. The correct term is dispersion.
- 9- Pg. 4 - Line 5 - 6000 rpm for 10 minutes. The centrifugal force should be provided. Also, the filtering method is not a good method to avoid contaminants, better to centrifuge, remove the solution, dispersed the slurry with an ultrasound bath and repeated this procedure at least 5 times.
- 10- Pg. 4 - Line 18 - Figure 1 should be removed; it is too basic for this level of journal.
- 11- Pg. 4 - Line 20 - The set consisted of the prepolymerized powder and the monomer liquid. The complete information about these Chemical should be given to allow any reader reproduction.
- 12- Pg. 5 - Line 48 - Figure 2. FE-SEM micrographs of fillers a) SiO_2 particles, b) FeAl-LDH and c) FeAl-LDH@ SiO_2 . Where is Figure 2C????
- 13- Pg. 5 - All the figures need to be replaced due to the low quality.
- 14- Pg. 5 - It is recommended that all the discussion come after each figure otherwise it is difficult to follow in the text. Use the topic results and discussion
- 15- Pg. 7 - Line 4 - related to the SEM images, better magnifications would better support the comments in the text.
- 16- Pg. 7 - Line 12 - The Fe/Al/Si ratio confirms that the majority of LDHs are found on the surface of silica particles. How the authors explain the composition of the EDS if the ratio LDH:Silica was of 1:1? This proportion is far in the analysis and no comments are provided in the text.
- 17- Pg. 8 - Line 19 - The XRD diffraction peaks should be indicated in interplanar distances after the position of the peaks in 2 theta and the basal distance calculated by the higher order peak possible, never the first one. The sizes of the crystalline domains can be calculated by Scherrer equation, the a and b cell parameters from the 110 peak. Which is intercalated anion?

18- Pg. 8 - Line 22 - In pattern FeAl-LDH@SiO₂ from figure 3 there are no characteristic peaks of LDH structure because FeAl-LDH was synthesized in the presence of amorphous SiO₂ and broad peaks of SiO₂ obscured them. This is not correct, if the proportion used is true, the XRD pattern of LDH should appear in the mixture. This can be an indication that a very small amount of LDH was deposited. This information should be given in the text by using proper analysis since EDS is not quantitative.

19- Pg. 8 - Line 24 - crystal structure remains non crystalline. This is really strange, if crystals exist, the material is crystalline.

20- Pg. 8 - Line 27 - no crystalline layer of LDH. This is unlikely to be true, particles were observed with layered morphology, consequently LDH must be in the sample and crystalline material.

21- Pg. 8 - Line 33 - The absorption peaks. FTIR spectra preset bands, not peaks. Please correct all the text.

22- Pg. 8 - Line 47 - The FeAl-LDH@SiO₂ particles are randomly dispersed in the PMMA matrix, as seen in images of the composite's fracture surface. This is really to see in the images, a higher magnification and mapping the elements would give better information in this sense.

Reviewer: 2

Comments to the Author(s)

The manuscript can be published after considering all the comments in the attached file.

Author's Response to Decision Letter for (RSOS-210835.R0)

See Appendix B.

RSOS-210835.R1 (Revision)

Review form: Reviewer 1

Is the manuscript scientifically sound in its present form?

Yes

Are the interpretations and conclusions justified by the results?

Yes

Is the language acceptable?

Yes

Do you have any ethical concerns with this paper?

No

Have you any concerns about statistical analyses in this paper?

No

Recommendation?

Accept with minor revision (please list in comments)

Comments to the Author(s)

Changes to be processes based on the previous recommendations:

2- Pg. 1 - Line 47 - exchangeable anions. Recently it was reported that LDH can also exchange cations and even cations/anions simultaneously, please update this information.

As it was requested, the information that LDH can only exchange anions is not correct. In 2019 a paper was published in the journal JACS indicating that LDH with a special composition can also exchange cations and even both simultaneously. This information needs to be included in the text.

4- Pg. 3 - Line 47 - Fe-Al LDH. A reason should be given for choosing this composition and also for the 3:1 molar ratio, which is very uncommon.

This is not the explanation why the authors decided to use the molar ration of 3:1. The most favorable molar ratio is 2:1 and not 3:1 of 4:1, please explain as requested.

9- Pg. 4 - Line 5 - 6000 rpm for 10 minutes. The centrifugal force should be provided. Also, the filtering method is not a good method to avoid contaminants, better to centrifuge, remove the solution, dispersed the slurry with an ultrasound bath and repeated this procedure at least 5 times.

The centrifuga force was included. Accepted or not by the authors, filtering is definitively not the best method to remove soluble impurities. There are a lot of references using filtration, but these references should be avoided since they are using an equivocal method of purification.

Authors are always using references to support the data, but they should have in mind that many papers are published with wrong data and these cannot be used as references to justify doubtfully results. When referencing use always references groups working in the area, being recognized as experts in the area and journals of high impact factors.

Review form: Reviewer 2 (Isa Emami Tabrizi)

Is the manuscript scientifically sound in its present form?

Yes

Are the interpretations and conclusions justified by the results?

Yes

Is the language acceptable?

Yes

Do you have any ethical concerns with this paper?

No

Have you any concerns about statistical analyses in this paper?

No

Recommendation?

Accept as is

Comments to the Author(s)

The authors have replied to all questions and comments. The current version of manuscript has been modified very well and it can be considered for publication as it is.

Decision letter (RSOS-210835.R1)

Dear Dr Vuksanovic:

Title: Inorganically modified particles FeAl-LDH@SiO₂ as reinforcement in PMMA matrix composite
Manuscript ID: RSOS-210835.R1

Thank you for submitting the above manuscript to Royal Society Open Science. On behalf of the Editors and the Royal Society of Chemistry, I am pleased to inform you that your manuscript will be accepted for publication in Royal Society Open Science subject to minor revision in accordance with the referee suggestions. Please find the reviewers' comments at the end of this email.

The reviewers and handling editors have recommended publication, but also suggest some minor revisions to your manuscript. Therefore, I invite you to respond to the comments and revise your manuscript.

Because the schedule for publication is very tight, it is a condition of publication that you submit the revised version of your manuscript before 19-Aug-2021. Please note that the revision deadline will expire at 00.00am on this date. If you do not think you will be able to meet this date please let me know immediately.

- 1) A text file of the manuscript (tex, txt, rtf, docx or doc), references, tables (including captions) and figure captions. Do not upload a PDF as your "Main Document".
- 2) A separate electronic file of each figure (EPS or print-quality PDF preferred (either format should be produced directly from original creation package), or original software format)
- 3) Included a 100 word media summary of your paper when requested at submission. Please ensure you have entered correct contact details (email, institution and telephone) in your user account
- 4) Included the raw data to support the claims made in your paper. You can either include your data as electronic supplementary material or upload to a repository and include the relevant doi within your manuscript
- 5) All supplementary materials accompanying an accepted article will be treated as in their final form. Note that the Royal Society will neither edit nor typeset supplementary material and it will

be hosted as provided. Please ensure that the supplementary material includes the paper details where possible (authors, article title, journal name).

Kind regards,
Dr Laura Smith
Publishing Editor, Journals

On behalf of the Subject Editor Professor Anthony Stace and the Associate Editor Professor Chaohua Cui.

RSC Associate Editor:
Comments to the Author:
(There are no comments.)

RSC Subject Editor:
Comments to the Author:
(There are no comments.)

Reviewer comments to Author:
Reviewer: 2
Comments to the Author(s)
The authors have replied to all questions and comments. The current version of manuscript has been modified very well and it can be considered for publication as it is.

Reviewer: 1
Comments to the Author(s)
Changes to be processed based on the previous recommendations:
2- Pg. 1 - Line 47 - exchangeable anions. Recently it was reported that LDH can also exchange cations and even cations/anions simultaneously, please update this information.

As it was requested, the information that LDH can only exchange anions is not correct. In 2019 a paper was published in the journal JACS indicating that LDH with a special composition can also exchange cations and even both simultaneously. This information needs to be included in the text.

4- Pg. 3 - Line 47 - Fe-Al LDH. A reason should be given for choosing this composition and also for the 3:1 molar ratio, which is very uncommon.

This is not the explanation why the authors decided to use the molar ratio of 3:1. The most favorable molar ratio is 2:1 and not 3:1 of 4:1, please explain as requested.

9- Pg. 4 - Line 5 - 6000 rpm for 10 minutes. The centrifugal force should be provided. Also, the filtering method is not a good method to avoid contaminants, better to centrifuge, remove the solution, dispersed the slurry with an ultrasound bath and repeated this procedure at least 5 times.

The centrifuga force was included. Accepted or not by the authors, filtering is definitively not the best method to remove soluble impurities. There are a lot of references using filtration, but these references should be avoided since they are using an equivocal method of purification.

Authors are always using references to support the data, but they should have in mind that many papers are published with wrong data and these cannot be used as references to justify doubtfully results. When referencing use always references groups working in the area, being recognized as experts in the area and journals of high impact factors.

Author's Response to Decision Letter for (RSOS-210835.R1)

See Appendix C.

Decision letter (RSOS-210835.R2)

Dear Dr Vuksanovic:

Title: Inorganically modified particles FeAl-LDH@SiO₂ as reinforcement in PMMA matrix composite

Manuscript ID: RSOS-210835.R2

It is a pleasure to accept your manuscript in its current form for publication in Royal Society Open Science. The chemistry content of Royal Society Open Science is published in collaboration with the Royal Society of Chemistry.

Yours sincerely,
Dr Ellis Wilde
Publishing Editor, Journals

On behalf of the Subject Editor Professor Anthony Stace and the Associate Editor Professor Chaohua Cui.

RSC Associate Editor
Comments to the Author:
(There are no comments.)

RSC Subject Editor
Comments to the Author:
(There are no comments.)

Reviewer(s)' Comments to Author:

Appendix A

This study investigates the effect of adding novel biobased fillers on mechanical properties of PMMA polymers. The investigators have presented results of materials characterization and mechanical assessment results to indicate the variations in behavior of composite specimens. The content of manuscript is interesting with very good potential for future research. However, the context of manuscript must be modified before publication and novelty of research should be signified more of authors. The comments of reviewer are given below. These minor comments must be resolved before publication.

General comments:

Increase the quality of images and check the fluency of the language in the context.

Specific comments:

Page 2-line 49: It will be better if authors explained why in LDH, the metal ions must have to possess an ionic radius similar to Mg^{2+} (0.65 Å)?

Page 3-line 1: what is the second role of the interlayer anions other than contributing to the expansion of the layered structure of LDH? The sentence seems to be incomplete?

Page 3-line 24 to 35: Authors must give more details about their state of art and importance of their research in the final paragraph of introduction.

Page 5-line 11-15: please revise the sentence about the measurement positions on each sample as you have mentioned twice that six positions were used for measurements.

Page 5-line 17: you have mentioned that test samples were standard. Which standard did you use?

Page 5-line 33: Revise the sentence.

Page 9-line 44-57: Please increase the quality of SEM images, since the details you have discussed are not clearly seen for reader.

Page 8 (last paragraph) and Page 9 (first paragraph) : you have reported dual values for the increase of hardness with addition of 3% and 5% FeAl-LDH@SiO₂ fillers. Please correct this ambiguity.

Page 9-line 7: You have specified that addition of fillers increases young modulus. Does it mean that addition of 50% filler materials will also result in extremely high elastic modulus? Is there an optimum number for addition of filler material to PMMA material.

The same question goes for hardness increase due to addition of filler to PMMA.

Page 9-line 15: The term "static loading experiments" does not seem to be applicable to your research since you have performed test by varying load over time with a constant speed of 0.5mm/min. Please clarify if you have done tests under quasi-static loading (constant increase of load over time) or fixed load over specific time intervals.

Page 9-line 16: what do you mean by ability of material to deform? Does it mean ductility? Failure strain?

Table 3: what do you mean by σ , K and ϵ ? Please define them in the text.

There is contradiction in authors' statements in last paragraph of section 5 (discussion). At the beginning they say at low deformation rates (tensile tests) the material deformability with addition of fillers decreases, on the other hand they state that at high strain rates (impact) material loses deformability with addition of filler material. So, at both loading scenarios the material loses deformability with increase of the filler content. There seems to be no material behavior varying with change of loading rate. And How is this issue similar to the behavior of thixotropic materials?

Appendix B

Dear reviewer,

The authors would like to thank the reviewer comments and suggestions. All the questions made us revising the paper and obtaining a more complete form of our work, and we are very grateful for your suggestions. In this document, the answers follow your remarks.

Reviewer's comments:

Reviewer: 1

The manuscript reports the purification of rice husk silica, deposition of LDH on the silica surface and use these materials as filler into PMMA. The topic is interesting and deserves publication but many corrections need to be implemented in the text before acceptance. Recommendations/changes/suggestions:

1- Pg. 1 – Line 46 - are lamellar compounds. Better to say: layered compounds

Answer: It is corrected in the text.

2- Pg. 1 – Line 47 - exchangeable anions. Recently it was reported that LDH can also exchange cations and even cations/anions simultaneously, please update this information.

Answer: In these hydroxide layers some of the M^{2+} cations are isomorphy substituted with M^{3+} thereby generating a positive charge. The higher (excess of) charge is compensated by the hydrated anions located in interlayer gallery. These layered configurations give them the ability to extensive use as anion-exchanging materials (adsorption, polymerization, catalysis, photochemistry, electrochemistry, biomedical and environmental application etc). LDH also has the ability of cation-exchange, which has been the subject of many researchers in recent years (photocatalysis...) [Parida K, Mohapatra L, Baliarsingh N. Effect of Co $2+$ Substitution in the Framework of Carbonate Intercalated Cu/Cr LDH on Structural, Electronic, Optical, and Photocatalytic Properties. J Phys Chem C [Internet]. 2012 Oct 25;116(42):22417–24. Available from: <https://pubs.acs.org/doi/10.1021/jp307353f>].

This was added to the text.

3- Pg. 1 – Line 50 - In LDH, the metal ions must have an ionic radius similar to Mg^{2+} (0.65 Å). The interlayer anions are carbonates, chlorides, nitrates and sulfates. The first information is not correct and the second is incomplete, organic and inorganic anions can be intercalated, please correct.

Answer: In LDH each metal cation is octahedrally coordinated by OH- and these OH- ions surrounding the space with radius of 0.07nm. Therefore, metal cations with radius not far from 0.07 nm can be incorporated into LDH. In the case of larger ions (Mn^{2+} , Pb^{2+} , Cd^{2+} , Ca^{2+} , Y^{3+} , La^{3+}) incorporation the close-stacking configuration will be distorting to some extent. The interlayer space is filled with exchangeable inorganic (carbonates, chlorides, nitrates, sulphates, complex anions etc.) or organic anions (carboxylates, alkyl sulfates, glycerolate, polymeric anions, biochemical anions etc.) along with water molecules [Zabihi O, Ahmadi M, Nikafshar S, Chandrakumar Preyeswary K, Naebe M. A technical review on epoxy-clay nanocomposites: Structure, properties, and their applications in fiber reinforced composites. Compos Part B Eng

[Internet]. 2018 Feb;135:1–24. Available from: <https://linkinghub.elsevier.com/retrieve/pii/S1359836817315342>].

This is added to the text in introduction.

4- Pg. 3 – Line 47 - f Fe-Al LDH. A reason should be given for choosing this composition and also for the 3:1 molar ratio, which is very uncommon.

Answer: The common ratios vary from 2 to 4 if the LDH structure is favored. Other ratios are producing other phases and sometimes lead to formation mostly of divalent ion oxides, in this case it would produce the oxide phase of ferrous ion. [M.V. Bukhtiyarova, A review on effect of synthesis conditions on the formation of layered double hydroxides, Journal of Solid-State Chemistry, Volume 269 (2019) 494-506].

Thus, this was the choice.

This is added to the introduction as explanation of the chosen ionic ratio

5- Pg. 3 – Line 48 - remove the pollution. Better to say: remove the contaminants.

Answer: It is corrected in the text.

6- Pg. 3 – Line 56 - were synthesized by a co-precipitation method. This information is incomplete, there are many different co-precipitations procedures, please explain details of the procedure.

Answer: FeAl-LDH (molar ratio Fe: Al = 3: 1) were synthesized by the method of co-precipitation from aqueous solutions under atmospheric conditions. FeCl₂·4H₂O (0.015 mol) and Al₂(OH)₅Cl·2.5 H₂O (0.005 mol) were dissolved in 100 ml of deionized water each. Silica particles were added to the glass beaker, which was placed on a magnetic stirrer, and aqueous solutions of FeCl₂·4H₂O and Al₂(OH)₅Cl·2.5 H₂O were prepared. The mass ratio of silica: LDH was 1:1. The aim of this synthesis is to form a precipitate layer (FeAl-LDH) on the present silica particles which are inert to the co-precipitation reaction. 1 mol/L NaOH was added dropwise to the solution until the pH reached 10 when the experiment was stopped. The dispersion was allowed to stand for 24 hours, then centrifuged at 6000 rpm for 10 minutes and then washed with water until the pH of the effluent solution was neutral. The solid with filter paper was dried at 80 °C for 24 hours to give FeAl-LDH@SiO₂ particles.

7- Pg. 3 – line 60 - 2.93 g of FeCl₂·4H₂O and 1.07 g of Al₂(OH)₅Cl·2,5 H₂O. In chemistry the amounts should be given in mols, not in masses. Please convert the numbers.

Answer: It is corrected in the text.

8- Pg. 4 – Line 4 - The suspension. The correct term is dispersion.

Answer: It is corrected in the text.

9- Pg. 4 – Line 5 - 6000 rpm for 10 minutes. The centrifugal force should be provided. Also, the filtering method is not a good method to avoid contaminants, better to centrifuge, remove the solution, dispersed the slurry with an ultrasound bath and repeated this procedure at least 5 times.

Answer: The centrifugal force is calculated by the formula

$$F_c = m \frac{v^2}{r}$$

Where v is the peripheral speed and r is the distance from the center of the centrifuge. The acceleration is then calculated as

$$a_c = \frac{4 \cdot \pi^2 \cdot r \cdot n^2}{60^2}$$

Which for the distance of the base of the test tube from the center of rotation of 6 cm, rotation speed of 6000 rpm gives the acceleration of 23663 m²/s which is 2366 times the gravity acceleration or 2366 g. In the text we added only the data 2366 times the g.

The filtration procedure was accepted from literature data as a standard one that enables the preparation of clean LDH particles after precipitation. [Zongxue Yu, Xiuhui Li, Yixin Peng, Xia Min, Di Yin and Liangyan Shao, MgAl-Layered-Double-Hydroxide/Sepiolite Composite Membrane for High-Performance Water Treatment Based on Layer-by-Layer Hierarchical Architectures, Polymers 2019, 11(3), 525; <https://doi.org/10.3390/polym11030525>]. The procedure you mentioned will be applied in further research and compared to the existing one.

10- Pg. 4 – Line 18 – Figure 1 should be removed; it is too basic for this level of journal.

Answer: It is corrected in the text. Figure 1 is removed from the text. Other images are renumbered.

11- Pg. 4 – Line 20 - The set consisted of the prepolymerized powder and the monomer liquid. The complete information about these Chemical should be given to allow any reader reproduction.

Answer: All denture base materials consist of two components one is the methyl methacrylate monomer and the powder is the polymerized PMMA. The powder that usually contains the initiator is dissolved in the monomer and after that the rest of the monomer polymerizes. This one is auto polymerizing material without cadmium which is required for the use in dentistry. The material is declared to be conform to the ISO 1567 class I type II.

12- Pg. 5 – Line 48 - Figure 2. FE-SEM micrographs of fillers a) SiO₂ particles, b) FeAl-LDH and c) FeAl-LDH@SiO₂. Where is Figure 2C????

Answer: It is corrected in the text.

13- Pg. 5 – All the figures need to be replaced due to the low quality.

Answer: All the figures provided are in 600 dpi and this is the best possible performance we can obtain from the SEM. We observed that in text downloaded from the site the quality was poor, so it seems to be the problem with image processing.

14- Pg. 5 – It is recommended that all the discussion come after each figure otherwise it is difficult to follow in the text. Use the topic results and discussion.

Answer: It is corrected in the text.

15- Pg. 7 – Line 4 – related to the SEM images, better magnifications would better support the comments in the text.

Answer: The best selected images are given and they represent the observed structure. The higher magnification induced the melting of the polymer matrix and this is why we could not obtain higher magnifications of composites structures.

16- Pg. 7 – Line 12 - The Fe/Al/Si ratio confirms that the majority of LDHs are found on the surface of silica particles. How the authors explain the composition of the EDS if the ratio LDH: Silica was of 1:1? This proportion is far in the analysis and no comments are provided in the text.

Answer: The deposition took place on the surface of the silica particles and most of the process was at that place. The EDS confirms the composition of the layer that was on the surface of silica particles. The excess of LDH was deposited probably in the space between particles and is visible as individual hexagonal particles on the image. That corresponds partly to some images of as deposited and formed LDH@SiO₂. [Xuqiang Ji 1, Wenling Zhang 1, Lei Shan 2, Yu Tian 2, Jingquan Liu, Self-assembly preparation of SiO₂@Ni-Al layered double hydroxide composites and their enhanced electrorheological characteristics, Sci Rep [Internet]. 2015 Dec;5:18367. Doi/ 10.1038/srep18367].

17- Pg. 8 – Line 19 - The XRD diffraction peaks should be indicated in interplanar distances after the position of the peaks in 2 theta and the basal distance calculated by the higher order peak possible, never the first one. The sizes of the crystalline domains can be calculated by Scherrer equation, the a and b cell parameters from the 110 peak. Which is intercalated anion?

Answer: The Scherrer formula could be applied only to the pure LDH particles as only there we are able to observe clear peaks resulting from crystal structure. In the LDH@SiO₂ structure the peaks are difficult to observe, as the layer of deposited LDF is very thin and the crystal structure could not be identified using this method, and only the result of the interaction of an amorphous phase is visible. The similar result was observed in the paper [Xuqiang Ji 1, Wenling Zhang 1, Lei Shan 2, Yu Tian 2, Jingquan Liu, Self-assembly preparation of SiO₂@Ni-Al layered double hydroxide composites and their enhanced electrorheological characteristics, Scientific Reports Volume 5 (2015) 18367. DOI: 10.1038/srep18367] where another type of LDH was deposited on silica particles having the similar structure as particles produced in this research. So, for particles used in this research this calculation could not be performed.

18- Pg. 8 – Line 22 - In pattern FeAl-LDH@SiO₂ from figure 3 there are no characteristic peaks of LDH structure because FeAl-LDH was synthesized in the presence of amorphous SiO₂ and broad peaks of SiO₂ obscured them. This is not correct, if the proportion used is true, the XRD pattern of LDH should appear in the mixture. This can be an indication that a very small amount of LDH was deposited. This information should be given in the text by using proper analysis since EDS is not quantitative.

Answer: Obviously, the EDS analysis indicate that the ratio of elements present on the surface correspond to the situation where the layer deposited on the surface is retaining the structure present in the precursor solution, and that the process didn't result in the deposition of all the LDH on the surface of the particles. This observation would be presented in the paper as follows:

The results from the EDS are suggesting that the layer of deposited LDH has the ratio of 3/1 of ions used in the preparation of the modifying layer. This crystal structure does not built the crystals that are easily recognizable from the XRD pattern. From available analysis it is possible to establish the layer of the inorganic material on the surface of the silica particles having predetermined ionic ratio. The structure was only observed using the SEM microscopy and some features are conform to features observed in [Xuqiang Ji 1, Wenling Zhang 1, Lei Shan 2, Yu Tian 2, Jingquan Liu, Self-assembly preparation of SiO₂@Ni-Al layered double hydroxide composites and their enhanced electrorheological characteristics, Scientific Reports Volume 5 (2015) 18367. DOI: 10.1038/srep18367] where better imaging technique was used. The ratio of Silica to LDH could be estimated from the EDS analysis and the result says that the atomic ratio could be considered as Si/Fe/Al=11.1/3/1.

19- Pg. 8 – Line 24 - crystal structure remains non crystalline. This is really strange, if crystals exist, the material is crystalline.

Answer: We rephrased this to material structure could be considered non crystalline.

20- Pg. 8 – Line 27 - no crystalline layer of LDH. This is unlikely to be true, particles were observed with layered morphology, consequently LDH must be in the sample and crystalline material.

Answer: The layer formed on the particles is very thin and does not produce the peaks in the diffraction pattern. This is where the conclusion came from. We reformulate this in the text. The same type of XRD pattern was observed in the [Xuqiang Ji 1, Wenling Zhang 1, Lei Shan 2, Yu Tian 2, Jingquan Liu, Self-assembly preparation of SiO₂@Ni-Al layered double hydroxide composites and their enhanced electrorheological characteristics, Sci Rep [Internet]. 2015 Dec;5:18367. Doi/ 10.1038/srep18367.].

21- Pg. 8 – Line 33 - The absorption peaks. FTIR spectra preset bands, not peaks. Please correct all the text.

Answer: It is corrected in the text.

22- Pg. 8 – Line 47 - The FeAl-LDH@SiO₂ particles are randomly dispersed in the PMMA matrix, as seen in images of the composite's fracture surface. This is really to see in the images, a higher magnification and mapping the elements would give better information in this sense.

Answer: Unfortunately elements mapping was not available to authors during this research. We confirm that that it would be ideal to examine and we will look forward to see if some opportunities in this direction would open during prospective studies. This is only the observational conclusion from the best magnification that was available to us.

Reviewer: 2

Page 2-line 49: It will be better if authors explained why in LDH, the metal ions must have to possess an ionic radius similar to Mg²⁺ (0.65 Å)?

Answer: In LDH each metal cation is octahedrally coordinated by OH⁻ and these OH⁻ ions surrounding the space with radius of 0.07nm. Therefore metal cations with radius not far from 0.07 nm can be incorporated into LDH.

This is corrected in the text

Page3-line 1: what is the second role of the interlayer anions other than contributing to the expansion of the layered structure of LDH? The sentence seems to be incomplete.

Answer: In LDHs, interlayer anions serve a dual purpose. First, they contribute to the LDHs' layered structure, increasing interlayer distance; on the other hand, they promote compatibility with the polymer matrix. It is added in the text.

Page 3-line 24 to 35: Authors must give more details about their state of art and importance of their research in the final paragraph of introduction.

Answer: The silica particles are a very well established reinforcement used in particle reinforced polymer composites. The interphase between the reinforcement and the matrix can improve material characteristics and often the silane molecules are added to the active places on silica particles in order to establish good bonding between phases. The idea for this modification was to modify the surface of silica particles using LDH that would further enable building of bonds with the matrix. This is also useful to modify the tendency of silica particles to agglomerate and to enable better dispersion of particles in the composite. The aim of the research was to examine the use of so modified particles and to improve mechanical properties of the obtained material. The mechanical properties are to be studied and commented in view of this sort of surface modification.

This is added in the introduction of the paper

Page 5-line 11-15: please revise the sentence about the measurement positions on each sample as you have mentioned twice that six positions were used for measurements.

Answer: the test is replaced and instead

During the measurements force, time and depth data were recorded. Because of the possibility of inhomogeneity in the sample, microindentation measurements were performed on six positions for each sample to obtain more results that are accurate and reduce the influence of possible inhomogeneity microindentation measurements were performed on six positions for each sample.

We replaced with

The measurement was done using the indenter having round tip with a diameter of 6mm. The machine enables the fine measurement of the force applied on the surface of the material and the measurements of the indentation depth, similar process as it is in nanoindentation. The process is repeated on six positions at the surface of the specimen and the data obtained are the mean of six measurements.

Page5-line 17: you have mentioned that test samples were standard. Which standard did you use?

Answer: ASTM International. ASTM E384-16, Standard Test Method for Micro indentation Hardness of Materials. West Conshohocken, PA. 2016. That reference is added in the text.

Page 5-line 33: Revise the sentence.

Answer: It is corrected in the text.

Figure 1 shows the morphology of synthesized silica particles and those with deposited LDH on the surface as revealed by FE-SEM analysis of the synthesized particles.

Page 9-line44-57: Please increase the quality of SEM images, since the details you have discussed are not clearly seen for reader.

Answer: The authors observed that the images are of poor quality when downloaded from the site so we suppose that the problem is in image processing. We changed the images for the correct visibility and hope that it will be better when it comes to you.

Page 8 (last paragraph) and Page 9 (first paragraph) : you have reported dual values for the increase of hardness with addition of 3% and 5% FeAl-LDH@SiO₂ fillers. Please correct this ambiguity.

Answer: Page 8 (last paragraph) shows the Vickers hardness, and page 9 (first paragraph talks about the hardness obtained by Micro indentation on Texture Analyzer EZ LX, Shimadzu. Additional clarification was added to the manuscript.

Page 9-line7: You have specified that addition of fillers increases young modulus. Does it mean that addition of 50% filler materials will also result in extremely high elastic modulus? Is there an optimum number for addition of filler material to PMMA material? The same question goes for hardness increase due to addition of filler to PMMA.

Answer: The addition of filler leads to an increase in hardness and modulus of elasticity, but the addition of 50 wt. % of filler does not lead to an extreme increase. With the increase of the filler content, agglomeration of the filler occurs and heavier mixing leads to a decrease in the elasticity modulus. 5 wt. % filler in our case is the optimal value to which the hardness and Young's modulus of elasticity increase. From the previous research the maximum improvement is reached before the content of the filler exceeds 10%. Usually optimal performance is obtained when the content is about 3%. At those amounts the structure profits from the interaction over interphase matrix-reinforcement. Larger quantities of reinforcement provide the possibility of particle - particle interaction and this is not beneficial for material mechanical properties.

Page 9-line 15: The term “static loading experiments” does not seem to be applicable to your research since you have performed test by varying load over time with a constant speed of 0.5mm/min. Please clarify if you have done tests under quasi-static loading (constant increase of load over time) or fixed load over specific time intervals.

Answer: This is actually the quasi-static loading of the material. We corrected this in the paper. The quasi static loading means that the load is increased over time with constant increase. The

real dynamic increase of the load is when the impact testing is performed. This is done in the impact testing machine where the speed of the impact is controlled.

Page 9-line 16: what do you mean by ability of material to deform? Does it mean ductility? Failure strain?

Answer: This is considered to be the maximum deformation during the tensile test.

Table 3: what do you mean by σ , K and ϵ ? Please define them in the text.

Answer: It is corrected in the text. Meanings are added below Table 3.

There is contradiction in authors' statements in last paragraph of section 5 (discussion). At the beginning they say at low deformation rates (tensile tests) the material deformability with addition of fillers decreases, on the other hand they state that at high strain rates (impact) material loses deformability with addition of filler material. So, at both loading scenarios the material loses deformability with increase of the filler content. There seems to be no material behavior varying with change of loading rate. And how is this issue similar to the behavior of thixotropic materials?

Answer: All the text in the paper is corrected and the statements are clarified. From the results it was obvious that the addition of the reinforcement in quasi static loading is decreasing the deformation of the specimen and increases the strength. If the energy absorbed during this test is calculated as the integration under the stress-strain curve it reveals that the energy decreases with addition of the reinforcement. This is in accordance with the literature for similar particulate reinforced polymer and this is cited in the paper. On the other hand when the specimen undergoes the high speed impact test the data about total energy that is transferred to the material is obtained in function to time. The total of the energy transferred to the material during the experiment increased with the addition of the reinforcements. The rate of load transfer to the material was proven to be the one of the most important parameters both for micro and nano composite materials and this confirms our observations. So in accordance to all of this the text in the discussion is changed to reflect those changes.

Appendix C

Dear Dr Smith,

We were very pleased receiving your answer and we reconsidered the remarks of the reviewer 2. We hope that the answers reflect our reconsideration of this exchange.

The answer about the characteristics of the LDH is added in the introduction. The main disagreement with the reviewer was about the preparation of particles, as LDH was a modifying layer on plant-based silica particles the procedure we used produced the material that was sensibly easier to admix in the preparation of the composite. The use of filtration in the production of reinforcement was a legitimate choice for us. We could use the centrifugation, rinse and ultrasonication in our future laboratory experience as the experience gained from this publication. Another point of disagreement was about the choice of molar ratio of Al/Fe. As the literature review says that it could range from 1 to 5, we considered that the ratio of 3 would give us the optimal deposit on the silica particle that will favor the formation of the desired structure on the surface and enable the improvement of material in terms of the mechanical properties.

We hope that those answers are satisfactory and we hope to continue contributing to your journal.

Sincerely

Marija Vuksanovic

Answer to reviewer 2

Dear Reviewer,

We are pleased that you invested your time and knowledge to improve the publication we are preparing for on the topic of use the LDH as the modifying layer on silica particles obtained from a plant. Your comments made our publication much better and made us reconsider our work and will influence further practices in our group. We made some changes to the manuscript and we are commenting them in this answer.

2- Pg. 1 – Line 47 - exchangeable anions. Recently it was reported that LDH can also exchange cations and even cations/anions simultaneously, please update this information. As it was requested, the information that LDH can only exchange anions is not correct. In 2019 a paper was published in the journal JACS indicating that LDH with a special composition can also exchange cations and even both simultaneously. This information needs to be included in the text.

According to the paper that appeared in JACS in 2019 the exchangeability of cations opens a new perspective of the use of LDH. We added this comment to the text and it is now as follows.

Answer: It was recently confirmed that LDH compounds, previously thought to be capable only to anion exchange, are able of cation exchange opening new perspective to use those materials (3). (Anne RS, Loana MB, Marco TG, and Fernando W, Cation Exchange Reactions in Layered Double Hydroxides Intercalated with Sulfate and Alkaline Cations

$(A(H_2O)_6)[M_2+6Al_3(OH)_18(SO_4)_2] \cdot 6H_2O$ ($M_{2+} = Mn, Mg, Zn$; $A_+ = Li, Na, K$), J. Am. Chem. Soc. [Internet] 2019, 141:531–540).

4- Pg. 3 – Line 47 - Fe-Al LDH. A reason should be given for choosing this composition and also for the 3:1 molar ratio, which is very uncommon.

This is not the explanation why the authors decided to use the molar ratio of 3:1. The most favorable molar ratio is 2:1 and not 3:1 or 4:1, please explain as requested.

According to the review paper (40) [Bukhtiyarova MV, A review on effect of synthesis conditions on the formation of layered double hydroxides, J Solid State Chem {Internet}. 2019;269:494-506] the ratios could vary from 1 to 5, the ratio of 2 to 4 leads to the formation of hydroxide structures and this was the aim of our intervention on the surface of silica particles. The intention was to get a layer of hydroxide on silica particles that would increase the bond between the material of reinforcement and that of the matrix. The text was modified in the part where the structure of the reinforcement is discussed as follows.

Answer:

Cations like Al^{3+} and Fe^{2+} were chosen as the most promising ones for use in reinforced composites. It was proven possible that by tailoring the chemical composition of reinforcement, mechanical properties improve significantly (18, 33). Among the main reasons for such behaviour is the compatibility with the polymer matrix due to different amounts of available surface hydroxide groups.

Research regarding the preparation of Fe(II)–Al layered double hydroxides revealed that ratios 2:1 (LDH-2) and 3:1 (LDH-3) gave materials with a similar crystalline structure (47). SEM analysis of the aforementioned LDH structures showed that LDH-3 possessed more compact structure/morphology than LDH-2. The loose structure of LDH-2 can be considered the most desirable one for the adsorption application since it is increasing the surface/contact area. But the structure of LDH-2 is not desirable for reinforced composites. The compact structure of reinforcement, such as LDH-3, will lead to improved mechanical properties of PMMA composites (46). In addition, a slightly higher amount of available hydroxyl groups of LDH-3 than LDH-2 observed in FTIR spectra indicates the possibility of establishment of better interconnection with the polymer matrix. Therefore, the ratio 3:1 was considered to be the optimal one for better compatibility with PMMA and the LDH-3 structure for emphasized reinforcing effect.

9- Pg. 4 – Line 5 - 6000 rpm for 10 minutes. The centrifugal force should be provided. Also, the filtering method is not a good method to avoid contaminants, better to centrifuge, remove the solution, dispersed the slurry with an ultrasound bath and repeated this procedure at least 5 times. The centrifuga force was included. Accepted or not by the authors, filtering is definitively not the best method to remove soluble impurities. There are a lot of references using filtration, but these references should be avoided since they are using an equivocal method of purification. Authors are always using references to support the data, but they should have in mind that many papers are published with wrong data and these cannot be used as references to justify doubtfully

results. When referencing use always references groups working in the area, being recognized as experts in the area and journals of high impact factors.

Filtration was an established way to rinse the LDH during preparation. This was the way we selected for the preparation of reinforcement based on literature review. The procedure to use centrifugation combined with ultrasonication could be applied too, but the particles we produced using the filtration behaved very well during composite preparation as the mixing of the particles with the monomer was much easier when this layer was deposited. We mention this in the part of preparation technique and now the text is as follows.

Response: We understand and appreciate the reviewer's concern. For feasible, fast, and easy industrial production, we have been looking for such a manner in LDH production. Thus, we have selected filtration as a first trial procedure in our research. Since we didn't have any interference during the composites manufacturing phase and the improvement of mechanical properties was achieved, we decided to select it for the batch production. Thus, we are reporting this study about the application of LDH particles in reinforced composites.

The text was modified to

The procedure was selected for the batch production of particles in this study for the simplicity and the the quantity of particles to be produced. The alternative procedure was using centrifugation and rinse of the product, and further ultrasonication and centrifugation could be alternative to the procedure used (3).